# Actin cytoskeleton deregulation confers midostaurin resistance in FLT3-mutant acute myeloid leukemia

Andoni Garitano-Trojaola[1], Ana Sancho [2,3], Ralph Götz[4], Patrick Eiring[4], Susanne Walz[5], Hardikkumar Jetani[1], Jesus Gil-Pulido[6], Matteo Claudio Da Via [1], Eva Teufel[1], Nadine Rhodes[1], Larissa Haertle [1], Estibaliz Arellano-Viera[1], Raoul Tibes[1], Andreas Rosenwald[7], Leo Rasche[1], Michael Hudecek [1], Markus Sauer [4], Jürgen Groll [2], Hermann Einsele [1], Sabrina Kraus[1,8] & Martin K. Kortüm [1,8✉]

The presence of FMS-like tyrosine kinase 3-internal tandem duplication (*FLT3*-ITD) is one of the most frequent mutations in acute myeloid leukemia (AML) and is associated with an unfavorable prognosis. FLT3 inhibitors, such as midostaurin, are used clinically but fail to entirely eradicate *FLT3*-ITD + AML. This study introduces a new perspective and highlights the impact of RAC1-dependent actin cytoskeleton remodeling on resistance to midostaurin in AML. RAC1 hyperactivation leads resistance via hyperphosphorylation of the positive regulator of actin polymerization N-WASP and antiapoptotic BCL-2. RAC1/N-WASP, through ARP2/3 complex activation, increases the number of actin filaments, cell stiffness and adhesion forces to mesenchymal stromal cells (MSCs) being identified as a biomarker of resistance. Midostaurin resistance can be overcome by a combination of midostaruin, the BCL-2 inhibitor venetoclax and the RAC1 inhibitor Eht1864 in midostaurin-resistant AML cell lines and primary samples, providing the first evidence of a potential new treatment approach to eradicate *FLT3*-ITD + AML.

[1] Department of Internal Medicine II, University Hospital Würzburg, Würzburg, Germany. [2] Department of Functional Materials in Medicine and Dentistry and Bavarian Polymer Institute, University of Würzburg, Würzburg, Germany. [3] Department of Automatic Control and Systems Engineering, University of the Basque Country UPV/EHU, San Sebastian, Spain. [4] Department of Biotechnology and Biophysics, Biocenter, University of Würzburg, Würzburg, Germany. [5] Comprehensive Cancer Center Mainfranken, Core Unit Bioinformatics, Biocenter, University of Wuerzburg, Wuerzburg, Germany. [6] Institute of Molecular Biology, Mainz, Germany. [7] Institute of Pathology, University of Würzburg, Würzburg, Germany. [8] These authors contributed equally: Sabrina Kraus, K. Martin Kortüm. ✉email: Kortuem_M@ukw.de

Acute myeloid leukemia (AML) is a genetically heterogeneous disease characterized by the clonal expansion of immature myeloid progenitor cells in the bone marrow (BM). Mutations of the FMS-like tyrosine kinase 3 (FLT3) gene occur in ~30% of AML cases, with internal tandem duplications (ITDs) being the most common type of mutation. These mutations lead to constitutive autophosphorylation of the FLT3 receptor; consequently, growth factor signaling pathways are activated. Thus, therapies that inhibit FLT3 kinase activity are highly warranted to disrupt disease progression. Drug discovery efforts have resulted in first- and second-generation FLT3 inhibitors, which can be classified as type I and type II based on their mechanism of interaction with the FLT3 receptor[1]. Type I inhibitors, including sunitinib (first-generation), midostaurin (first-generation), lestaurtinib (first-generation), crenolanib (second-generation), and gilteritinib (second-generation) interact with the ATP-binding site when the receptor is in the active conformation, while type II inhibitors, such as sorafenib (first-generation), ponatinib (first-generation), and quizartinib (second-generation), interact with a hydrophobic region directly adjacent to the ATP-binding domain that is accessible only when the receptor is inactive and prevent receptor activation. Unlike type II inhibitors, type I inhibitors reduce FLT3 phosphorylation in FLT3-ITD- and FLT3-TKD-mutated AML cells, while type II inhibitors can target FLT3-ITD but lack efficacy against TKD mutations, which exhibit secondary mechanism resistance to type II inhibitors[1–3]. Gilteritinib and midostaurin have been approved by the US Federal Drug Administration (FDA) to treat individuals with FLT3-mutant AML. Nevertheless, FLT3-ITD AML is associated with unfavorable prognosis, and patients develop drug resistance, with the underlying mechanisms remaining largely unexplained[4,5].

FLT3 receptor glycosylation and its autophosphorylation have been associated with resistance to FLT3 inhibitors in FLT3-ITD AML[6–9]. The exogenous expression of FLT3-ITD or FLT3-ITD mutations leads to Ras-related C3 botulinum toxin substrate 1 (RAC1) activation[10]. RAC1 belongs to the RHO GTPase family and is involved in many cellular functions, such as actin polymerization, cytoskeletal organization, migration, and apoptosis[11]. The polymerization of actin monomers (known as G-actin) is mainly carried out in four steps: (1) RAC1-GTP and phosphatidylinositol 4,5-bisphosphate (regulated positively by RAC1) bind to WAVE2 and N-WASP; (2) WAVE2 and N-WASP undergo conformational changes to an activated state; (3) the activated forms of N-WASP and WAVE2 recruit the ARP2/3 complex; and (4) the multiprotein complex comprising RAC1 + P-WAVE2 or P-N-WASP + ARP2/3 initializes actin polymerization to form actin filaments (known as F-actin)[11–13]. Moreover, profilin-1 (PFN1), which binds to N-WASP or WAVE proteins, can accelerate actin polymerization[14]. Deregulation of this process through increasing the number of actin filaments and cell stiffness enhances the invasiveness and therapy resistance in solid tumor[15,16]. Moreover, RAC1 inhibits apoptosis by binding to and stabilizing antiapoptotic BCL-2 family proteins (BCL-2 and MCL1) in colon cancer, cervical cancer, and B-cell lymphoma[17–19].

We hypothesized that FLT3 receptor phosphorylation and glycosylation, which have been associated with FLT3 inhibitor resistance, may induce RAC1 hyperactivation, which consequently deregulates of actin dynamics and the antiapoptotic BCL-2 family and may confer midostaurin resistance in FLT3-mutant AML.

## Results

**FLT3-dependent RAC1 hyperactivation in FLT3-ITD + midostaurin-resistant cells**. To study the role of RAC1-regulated cellular processes in response to FLT3 inhibitors, we developed midostaurin-resistant AML cell lines from the FLT3-ITD homozygous and heterozygous AML cell lines MV4-11 and MOLM-13, respectively. Parental cells are designated MID-Sens, and midostaurin-resistant cells are designated MID-Res. The IC50 of midostaurin in MV4-11 cells increased from 15.09 to 55.24 nM in MID-Sens and MID-Res cells; that of MOLM-13 cells increased from 29.41 to 87.83 nM (Fig. 1a). Afterwards, we characterized FLT3 receptor glycosylation and phosphorylation by western blotting, flow cytometry and direct stochastic optical reconstruction microscopy (dSTORM). In MV4-11 and MOLM-13 MID-Res cells, FLT3 receptor phosphorylation at tyrosine 969 (residue) was upregulated compared to that in their respective MID-Sens cell lines (Figs. 1b and S5). Additionally, FLT3 receptor glycosylation was upregulated in MV4-11 MID-Res cells versus MV4-11 MID-Sens cells (Figs. 1b and S5). To determine the impact of midostaurin on FLT3 receptor activity, MV4-11 and MOLM-13 MID-Sens and MID-Res cells were treated with 50 nM midostaurin for 24 h. A single cycle of midostaurin treatment increased FLT3 receptor glycosylation in MID-Sens and MID-Res cells. However, the reduction in tyrosine 969 phosphorylation was more pronounced in MID-Sens than in MID-Res cells (Figs. 1b and S5), with MID-Res cells showing no reduction in FLT3 phosphorylation. The FLT3 glycosylation level was correlated with FLT3 surface expression. As expected, FLT3 surface expression as measured by flow cytometry was increased in MV4-11 and MOLM-13 MID-Res cells compared with the respective MID-Sens cells (Fig. 1c). Moreover, dSTORM revealed the existence of FLT3 clusters in the plasma membrane of MID-Res cells, which was not observed in MID-Sens cells (Fig. 1d).

Next, we quantified RAC1 activation (RAC1-GTP) in our two model cell lines: RAC1-GTP levels were increased in MV4-11 and MOLM-13 MID-Res cells compared to their respective MID-Sens cells (Fig. 1e). A single dose of midostaurin significantly decreased RAC1 activation in MV4-11 and MOLM-13 MID-Sens cells (Fig. 1e), but RAC1 activation was not reduced by 50 nM midostaurin in MID-Res cells (Fig. 1e). To show that FLT3 was responsible for RAC1 hyperactivation in MV4-11 and MOLM-13 MID-Res cells, we performed siRNA knockdown (KD) of FLT3 (Figs. 1f). FLT3 KD significantly reduced RAC1 activation in MV4-11 and MOLM-13 MID-Res cells (Fig. 1f). Moreover, FLT3 KD reversed midostaurin resistance in MV4-11 and MOLM-13 MID-Res cells (Fig. S1A).

**The RAC1 inhibitor Eht1864 overcomes midostaurin resistance by inducing cell cycle arrest in G1 and apoptosis in MID-Res cell lines and primary samples**. To define the functional role of RAC1 activation in resistance to midostaurin, we treated cells with the specific RAC1 inhibitor Eht1864. Eht1864 binds to RAC1 and keeps it in an inert and inactive RAC1 state, which prevents GEF-mediated nucleotide exchange as well as RAC1 binding to downstream effectors[20]. In our experiments, Eht1864 in combination with midostaurin synergistically reduced cell viability in MV4-11 and MOLM-13 MID-Res cells (Fig. 2a). This combination induced cell cycle arrest in G1 phase after 24 h of treatment (Fig. 2b). Subsequently, apoptosis activation and cell death induction in MID-Res cell lines were observed by caspase-3 cleavage protein and annexin V/PI assays upon treatment with this combination for 24 and 48 h, respectively (Figs. 2c).

Once we confirmed that RAC1 inhibition overcomes midostaurin resistance, we explored the effect of this combination in cells obtained from patients with refractory FLT3-ITD+ and FLT3-TKD + AML. A 24-h treatment with Eht1864 in combination with midostaurin synergistically reduced the viability of relapsed cells by inducing cell death (Fig. 2d, e). In addition, this

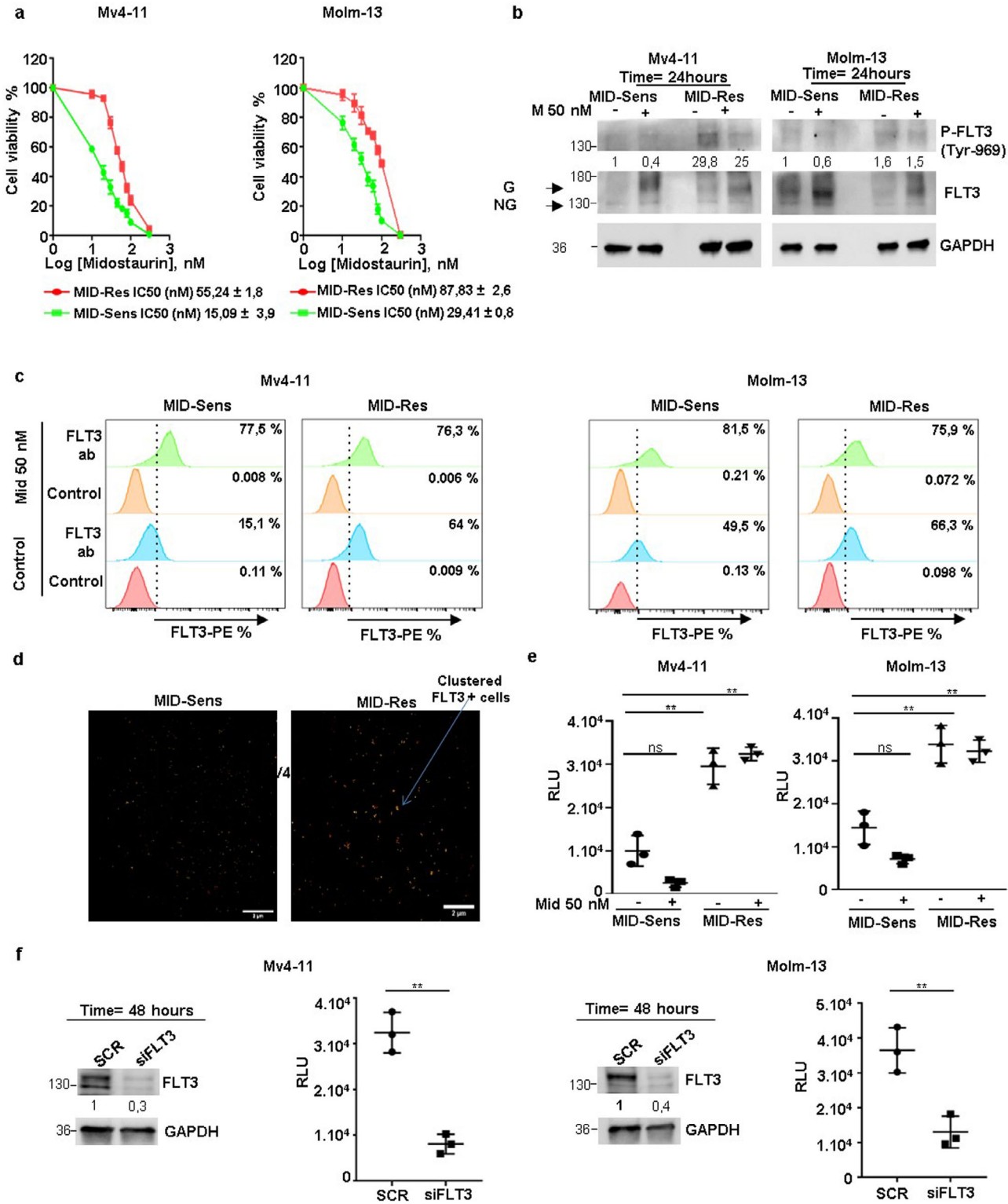

combination did not induce toxicity in peripheral blood mononuclear cells (PBMCs) obtained from healthy donors (Fig. 2e).

**Midostaurin-dependent actin polymerization machinery is overexpressed in midostaurin-resistant cells.** To understand the underlying mechanisms of RAC1-induced midostaurin resistance, we focused on the genes that regulate actin polymerization. We found that the protein levels of total and phosphorylated N-WASP, WAVE2, and PFN1 were enhanced in MV4-11 and

MOLM-13 MID-Res cells compared to their respective MID-Sens cells (Fig. 3a). Additionally, the expression of WASP protein (48% homology to N-WASP), which possesses lower actin polymerization activity than N-WASP, was upregulated in MV4-11 and MOLM-13 MID-Res versus their respective MID-Sens cells (Figs. 3a and S6). To determine the impact of midostaurin on the regulation of the actin polymerization machinery, MV4-11 and MOLM-13 MID-Sens and MID-Res cells were treated for 24 h with 50 nM midostaurin. A single treatment cycle of midostaurin

**Fig. 1 FLT3 dependent RAC1 hyperactivation in MV4-11/MOLM-13 Midostaurin-resistant cells. a** Midostaurin (Mid) IC50/cell viability and FLT3 glycosylation/phosphorylation analysis in MV4-11/MOLM-13 Mid sensitive (MID-Sens) and resistant (MID-Res) cells. **b** FLT3 phosphorylation/ glycosylation was analyzed in MV4-11/MOLM-13 MID-Sens ± MID and MID Res ± MID by Western blot. FLT3 blot, G means glycosylated band (160 kDa) and NG, non glycosylated band (130 kDa). **c** FLT3 surface expression was analyzed in MV4-11/MOLM-13 MID-Sens ± MID and MID Res ± MID by flow cytometry. **d** FLT3 receptor was analyzed in MV4-11 MID-Res versus MID-Sens by super-resolution microscopy dSTORM. **e** RAC1 activation was analyzed in MV4-11/MOLM-13 MID-Res ± Mid 50 nM compared to MID-Sens cells ± Mid 50 nM (MID-Sens + Mid 50 nM versus MID-Sens − Mid (MV) $p =$ 0.059, $t = 3.4$, df = 2.3 (MO) $p = 0.07$, $t = 3.1$, df = 2.3; MID-Res − Mid versus MID-Sens − Mid (MV) $p < 0.01$, $t = 6.1$, df = 3.8 (MO) $p < 0.01$, $t = 5.7$, df = 3.9; MID-Res + Mid versus MID-Sens (MV) $p < 0.01$, $t = 10.03$, df = 2.7 (MO) $p < 0.01$, $t = 6.5$, df = 3.4). **f** RAC1 activation was analyzed in FLT3 KD MV4-11/MOLM-13 MID-Res compared to scramble MV4-11 MID-Res cells ((MV) $p < 0.01$, $t = 8.56$, df = 2.9, (MO) $p < 0.01$, $t = 5.1$, df = 3.5). Relative light units (RLU). The western blot results are normalized by loading control (GAPDH) and are expressed as fold change relative to the control. Data are shown as means ± SDs (error bars) from three independent experiments.

reduced WAVE2 phosphorylation but not N-WASP phosphorylation in MV4-11 and MOLM-13 MID-Res cells (Figs. 3a and S6). Treatment with FLT3 KD and the RAC1 inhibitor Eht1864 reversed midostaurin resistance, reduced N-WASP phosphorylation in MV4-11 and MOLM-13 MID-Res cells (Figs. 3b). To determine whether the midostaurin response was regulated via actin polymerization, WAVE2, N-WASP, WASP, the ARP2/3 complex, and PFN1 were specifically targeted by siRNAs. The inhibition of N-WASP, the ARP2/3 complex, PFN1 and WASP reduced the proliferation of MV4-11 and MOLM-13 MID-Res cells over four days (Figs. 3c and S2A). However, WAVE2 KD did not induce any change in the proliferation of MV4-11 and MOLM-13 MID-Res cells (Fig. 3c). The IC50 of midostaurin was significantly reduced in MV4-11 and MOLM-13 MID-Res cells with N-WASP/ARP2/PFN1 KD compared to cells treated with scramble (Fig. 3c). No changes in the IC50 value of midostaurin were observed in MV4-11 and MOLM-13 MID-Res cells with WAVE2 and WASP KD (Figs. 3c, d and S2A). The decrease in cell viability observed upon N-WASP/ARP2/3/PFN1 inhibition was due to cell death induction, and this decrease was enhanced by midostaurin treatment (Fig. S2B). Additionally, the actin cytoskeleton disruptor Cytochalasin D and ARP2/3 small-molecule inhibitor CM-636 reversed midostaurin resistance in MID-Res cells via cell death induction (Fig. S2C).

To determine whether this regulation occurs in AML patients, correlation analysis between the expression of actin polymerization positive regulators (N-WASP, WAVE2, ARP2/3 complex, PFN1) and FLT3 signaling genes was performed by using publicly available microarray expression data (E-MTAB-3444). A positive correlation between the expression of these four inducers of actin polymerization and FLT3 signaling genes was observed in patients with *FLT3*-ITD ($r = 0.67$, $n = 178$) and FLT3 WT ($r = 0.57$, $n = 461$) de novo AML (Fig. 3e). Correlating the expression of each activator of actin polymerization individually with FLT3 pathway gene expression showed a moderate positive correlation for WAVE2/PFN1/ARP2 in *FLT3*-ITD and FLT3 WT de novo AML patients (Fig. S3A–C), but no correlation was observed between N-WASP and FLT3 pathway gene expression (Fig. S3D).

**RAC1 hyperactivation increases cell stiffness and enhances adhesion forces to mesenchymal stromal cells in midostaurin-resistant *FLT3*-ITD+ AML cells.** ARP2/3 activation is essential in the induction of actin polymerization[21]. The grade of actin polymerization positively correlates with actin filament growth, which is an important regulator of cell mechanical properties such as cell stiffness[22–24]. To show that the ARP2/3 complex was overactivated in MV4-11 and MOLM-13 MID-Res cells, we analyzed the actin filament load and stiffness in MV4-11 and MOLM-13 MID-Sens and MID-Res cells. We first visualized the load of actin filaments by high-resolution structured illumination microscopy (SIM) and observed an increased number of actin filaments in MV4-11/MOLM-13 MID-Res cells compared to their

respective MID-Sens cells (Fig. 4a). Then, MV4-11 and MOLM-13 MID-Res cells were treated with the combination of Eht1864 and midostaurin for 24 h, and we observed a drastic reduction in the number of actin filaments in MV4-11 and MOLM-13 MID-Res cells (Fig. 4a). To assess changes in cell stiffness, we applied FluidFM®, a new microfluidic technology adapted to traditional atomic force microscopy (AFM)[25], which confirmed significant increases in cell stiffness in MV4-11 and MOLM-13 MID-Res cells compared to MID-Sens cells (Fig. 4b). Likewise, the combination of Eht1864 and midostaurin for 24 h reduced cell stiffness in MV4-11 and MOLM-13 MID-Res cells (Fig. 4b). Moreover, as previously described by others, RAC1 activation increases the adhesion forces of cancer cells to mesenchymal stromal cells (MSCs), which negatively impacts the therapeutic response[26]. Accordingly, we independently co-cultured MID-Sens and MID-Res cells with the MSC line HS-5 for 24 h. The adhesion forces of MV4-11 MID-Res cells to HS-5 cells as well as those of MID-Sens cells to HS-5 cells were assessed by FluidFM®, and MV4-11 MID-Res cells were attached more strongly to HS-5 cells than were the MID-Sens cells (Fig. 4c and Supplementary Movies 1–2). Strikingly, the combination of Eht1864 and midostaurin (incubated for 24 h) was able to detach MV4-11 MID-Res cells from the MSCs (Supplementary Movies 3–5).

**The combination of Eht1864 and venetoclax reverses midostaurin resistance by blocking the antiapoptotic axis BCL-2/ MCL1.** As RAC1 is a positive regulator of BCL-2/MCL1, we studied the role of the BCL-2 family in the response to midostaurin in FLT3-mutant AML. BCL-2 and MCL1 expression was strongly upregulated in MV4-11 MID-Res cells compared to their parental cells (Figs. 5a); no changes were observed in MOLM-13 cells. To determine the role of midostaurin in BCL-2/MCL1 regulation, MV4-11 and MOLM-13 MID-Sens and MID-Res cells were treated with 50 nM midostaurin for 24 h. MCL1 expression was decreased by midostaurin in MV4-11 and MOLM-13 MID-Sens and MID-Res cells (Figs. 5a); however, BCL-2 expression was increased by midostaurin in MID-Sens cells (Figs. 5a). In addition, MV4-11 and MOLM-13 MID-Res cells were more sensitive to the BCL-2 inhibitor venetoclax than were the respective MID-Sens cell lines (Fig. 5b). Notably, the combination of midostaurin and Eht1864 decreased MCL1 expression but not BCL-2 expression in MV4-11 and MOLM-13 MID-Res cells (Figs. 5c). BCL-2 in the absence of MCL1 sequesters the proapoptotic proteins BIM/BAX/BAK and partially inhibits apoptosis[27]. Thus, BCL-2 expression could induce secondary resistance to the combination of midostaurin and Eht1864 in AML. Thus, we hypothesized that venetoclax would exhibit synergistic effects of with the midostaurin/Eht1864 combination therapy. Notably, venetoclax in combination with either midostaurin or Eht1864 synergistically reduced the viability of MV4-11 and MOLM-13 MID-Res cells by significantly abrogating MCL1/BCL-2 expression and inducing cell death (Figs. 5d, e and

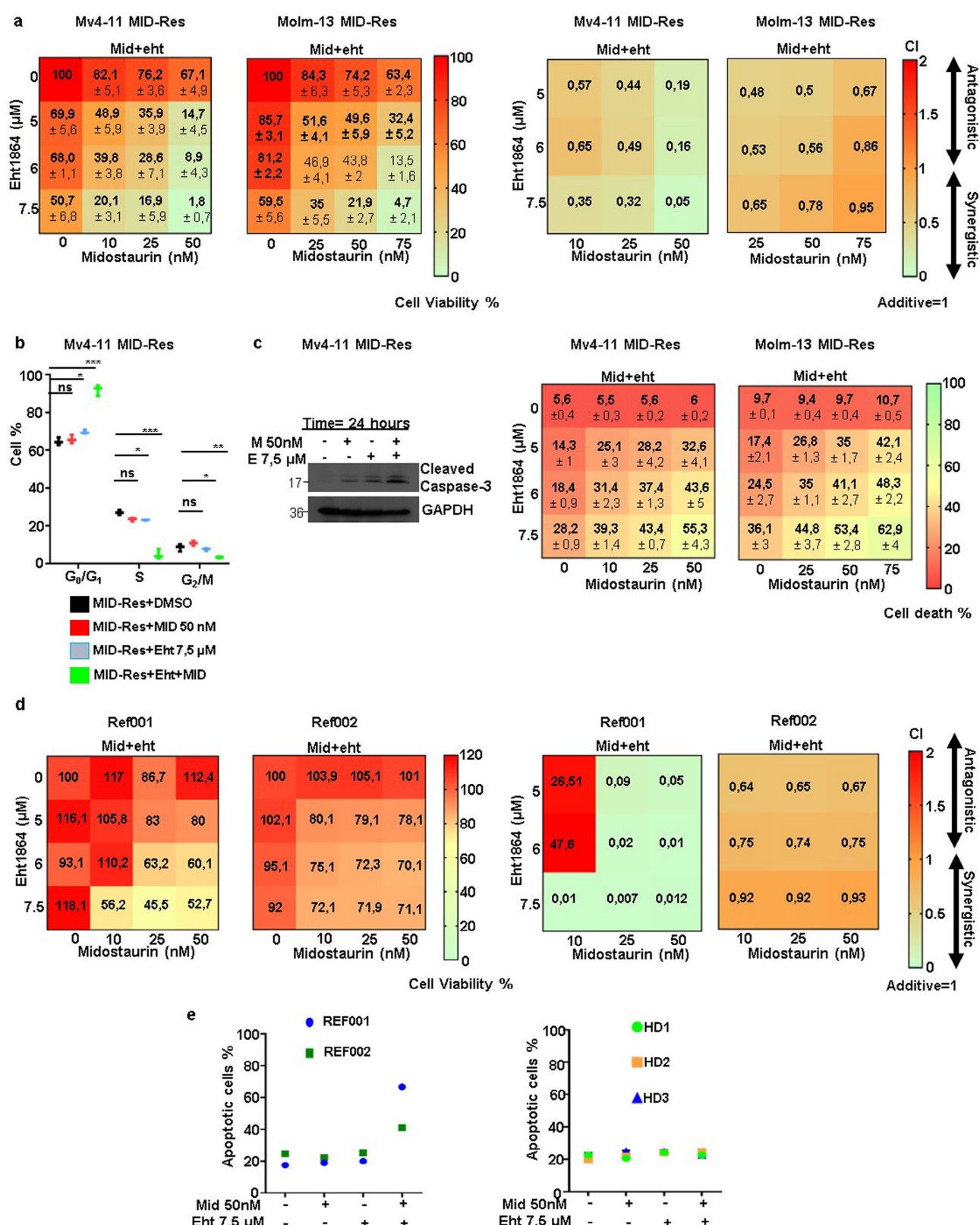

S4A, B). The results were confirmed in primary samples from two patients with refractory *FLT3*-ITD AML; cell death was induced in the majority of cells (>65%) (Fig. 5f). Of note, this triple combination was not toxic in PBMCs obtained from three healthy donors (Fig. 5f).

## Discussion

Activation of the PI3K and JAK/STAT pathways is a known resistance mechanism against FLT3 inhibition in *FLT3*-ITD AML[28–31]. RAC1 is a positive regulator of both pathways and interacts with FLT3[10,32]. This cross talk mediates drug resistance

**Fig. 2 RAC1 inhibitor Eht1864 overcomes Midostaurin resistance by the induction of cell cycle arrest in G1 and apoptosis in MID-Res cell lines as well as primary samples. a** The cell viability and synergy analysis of Mid + Eht1864 (Eht) combination was done in MV4-11/MOLM-13 MID-Res treated with these compounds during 48 h. **b** The cell cycle analysis was carried out in MV4-11 MID-Res treated with Mid and Eht alone/combination during 24 h. **c** The apoptosis was studied by caspase-3 cleavage monoclonal antibody and cell death % was defined by Annexin V + /PI− + Annexin V + PI+ in MV4-11/MOLM-13 MID-Res cells treated with Mid and Eht alone/combination during 24 and 48 h, respectively. **d** Cell viability and synergy analysis of Eht + Mid combination was done in Refractory (REF) REF001/REF002 samples treated with these compounds during 48 h. **e** The apoptosis analysis in REF001/REF002 and in PBMCs obtained from healthy donors treated with Mid and Eht alone/combination during 24 h. Data are shown as means ± SDs (error bars) from three independent experiments.

against classical chemotherapy via regulation of DNA repair pathways[10,19]. However, the role of RAC1 in the efficacy of FLT3 inhibitors in the treatment of *FLT3*-ITD leukemia has not yet been determined. Our study provides the first evidence that RAC1-dependent actin cytoskeleton remodeling and inhibition of apoptosis may underlie midostaurin resistance in *FLT3*-ITD AML.

*FLT3*-ITD mutations, which increase FLT3 autophosphorylation, lead to the activation of RAC1 in AML[10,33]. Clinical responses to FLT3 inhibitors have been correlated with the inhibition of FLT3 phosphorylation[7]. Moreover, FLT3 receptor glycosylation that is induced by FLT3 inhibitors, has been associated with FLT3 inhibitor resistance[9]. Accordingly, we observed increases in FLT3 phosphorylation/glycosylation and RAC1 hyperactivation in MV4-11 and MOLM-13 cells resistant to midostaurin, which was reversed by specifically targeting the FLT3 receptor. We used Eht1864, a specific RAC1 inhibitor, to investigate the effects of its hyperactivated state on midostaurin resistance. This compound is 20-fold more potent than other RAC1 inhibitors (Nsc23766) used in other AML studies[10]. Eht1864 reversed midostaurin resistance by inducing cell cycle arrest in G1 and activating apoptosis in established *FLT3*-ITD AML cell lines and in primary cells obtained from refractory AML patients with *FLT3*-ITD and *FLT3*-TKD D835 mutations. Weisberg et al. demonstrated that midostaurin reduces the viability of FLT3-mutated Ba/F3 leukemia cell lines and mouse models by inducing G1 arrest and apoptosis[6]. Therefore, this study corroborates our result that RAC1 inhibition overcomes the antitumor activity of midostaurin in FLT3-mutant clones resistant to midostaurin. Of note, Eht1864 and midostaurin either alone or in combination did not show any toxicity in PBMCs obtained from healthy donors. These results are in accordance with other studies, where no cell death was observed in healthy cells treated with these compounds [17,34].

RAC1 is a master regulator of actin cytoskeleton rearrangement[35]. RAC1 is activated by binding to N-WASP and WAVE2, which induces the conformational activation of these two proteins. Then, N-WASP and WAVE2 can bind to the ARP2/3 complex and activate it, resulting in actin polymerization[36,37]. The phosphorylation of N-WASP and WAVE2 is positively correlated with binding to the ARP2/3 complex and actin polymerization activation[36,37]. Any dysregulation in actin polymerization regulators favors cancer cell proliferation, migration, metastases and drug resistance in solid tumor[15,38]. However, their role in treating hematological malignancies has not yet been described. Here, we provide evidence that in *FLT3*-ITD AML, RAC1-dependent actin cytoskeleton remodeling plays a substantial role in the acquisition of resistance to midostaurin in vitro. N-WASP and WAVE2 phosphorylation is upregulated in midostaurin-resistant cells. N-WASP-specific KD but not WAVE2 KD reversed midostaurin resistance. Moreover, ARP2/3 complex inhibition by the small-molecule inhibitor CK-636 and siRNAs restored midostaurin activity. These results show the importance of N-WASP phosphorylation and ARP2/3 complex activation in midostaurin resistance in FLT3-mutant AML. Regarding WAVE2, because midostaurin is able to reduce its phosphorylation, we think that WAVE2 is not critical for midostaurin resistance in AML. In

addition, proteins that stimulate actin polymerization, such as PFN1, have been associated with resistance to the proteasome inhibitor bortezomib in multiple myeloma[39]. Additionally, we demonstrated that PFN1 expression is correlated with midostaurin resistance in FLT3-mutant AML cells.

Overactivation of actin polymerization leads to increased production of actin filaments[40]. As expected, MV4-11 and MOLM-13 midostaurin-resistant cells showed an increase in the number of actin filaments and cell stiffness compared to their respective midostaurin-sensitive parental cells. In line with our results, recent studies have described that tumor stiffening promotes resistance to the tyrosine kinase inhibitor sorafenib in triple-negative breast cancer cells and that cell rigidity is higher in cisplatin-/paclitaxel-resistant ovarian cancer cells versus parental ovarian cancer cells[41–43]. Moreover, massive actin filament accumulation has been associated with resistance to natural killer cell-based immunotherapy and chemotherapy in ovarian and breast cancers, respectively[43,44]. These results suggest that the cytoskeletal architecture and physical properties of cancer cells could play a critical role in drug resistance and could be novel parameters to evaluate therapy response.

It has been hypothesized that remodeling of the actin cytoskeleton is a strategy by which tumor cells evade apoptosis[45]. In fact, RAC1 blocks apoptosis by stabilizing the antiapoptotic proteins BCL-2 and MCL1 and upregulating their expression in cancer[17,18]. In line with that, MV4-11 MID-Res cells showed higher BCL-2 and MCL1 expression than did the parental cells. A single dose of midostaurin decreased MCL1 expression in sensitive and resistant cell lines. However, BCL-2 expression was upregulated under midostaurin treatment. As RAC1 activation is reduced under Midotaurin treatment in MV4-11 and MOLM-13 parental cell lines, RAC1 might regulate MCL-1 stability but not BCL2. Further investigation is needed to elucidate the role of RAC1 in the stability of BCL2/MCL-1 in MID-Sens/MID-Res cells. Due to BCL2 overexpression, higher sensitivity to the BCL-2 inhibitor venetoclax was observed in resistant cells than in sensitive cells. Recently, Ma et al. described that FLT3 inhibitors such as midostaurin and gilteritinib sensitize *FLT3*-ITD de novo AML cells to venetoclax by reducing MCL1 expression[34]. To induce full apoptosis in cancer cells, inhibition of MCL1 and BCL-2 is needed[46]. The combination of midostaurin and Eht1864 degrades MCL1 but increases BCL-2 expression. BCL-2 sequesters proapoptotic genes such as BIM and partially inhibits apoptosis[47]. Therefore, we believe that BCL-2 expression may contribute to secondary resistance to Eht1864 and midostaurin combination therapy. Consequently, we decided to add venetoclax, which was recently approved by the FDA in combination with hypomethylating agents or low-dose cytarabine for elderly individuals (75 years or older) with newly diagnosed AML or those who have comorbidities precluding intensive induction chemotherapy, to investigate whether BCL-2 inhibition enhanced the antitumor activity of the Eht1864/midostaurin combination therapy in FLT3-mutant AML. Strikingly, this triple combination comprising Eht1864, venetoclax and midostaurin synergistically reduced the viability of *FLT3*-ITD midostaurin-resistant cells and cells

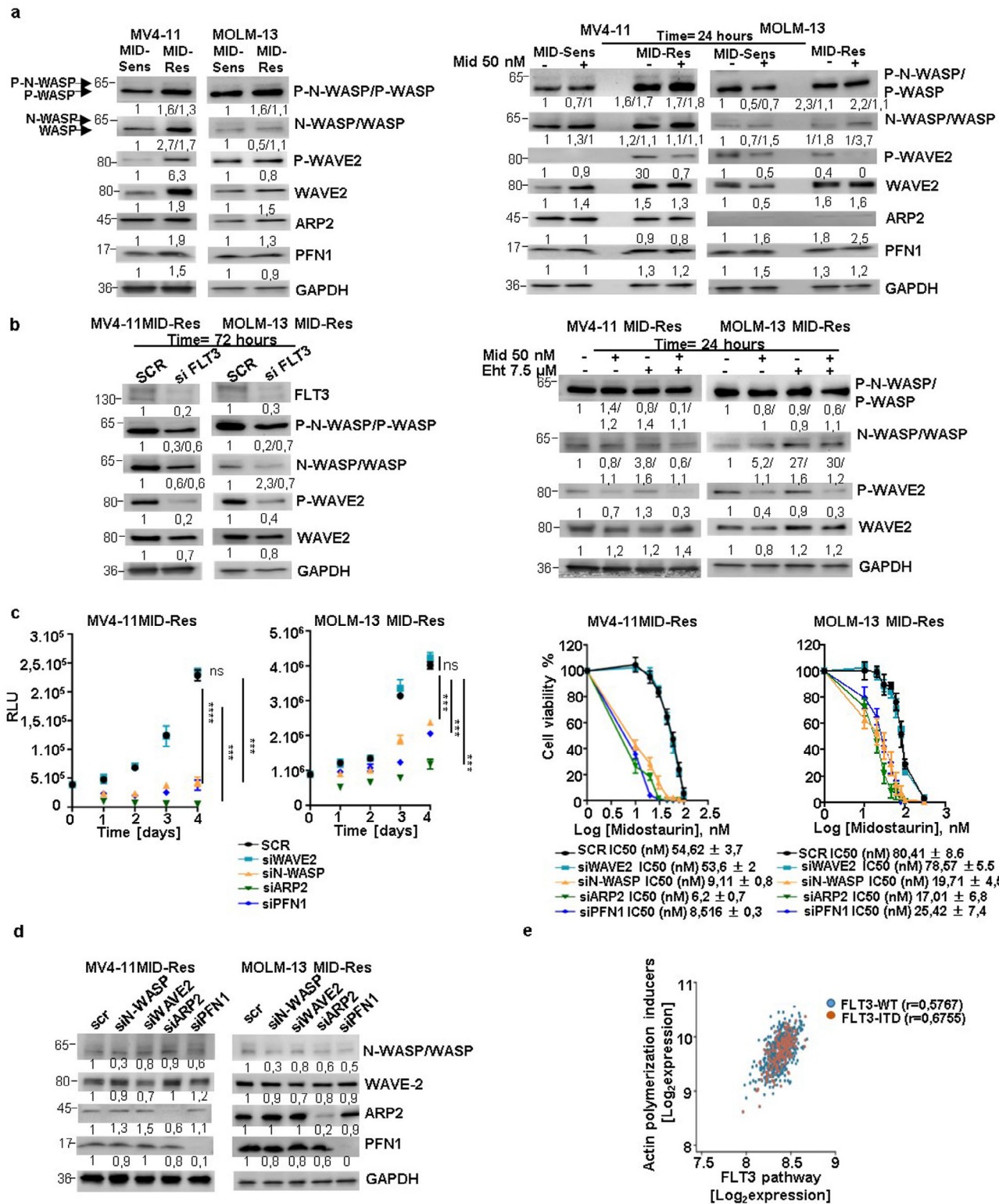

obtained from patients with *FLT3*-ITD refractory AML by reducing MCL1 and BCL-2 expression. Notably, no apoptosis was observed in PBMCs obtained from healthy donors treated with the triple combination therapy.

In summary, we propose a new midostaurin resistance mechanism in FLT3-mutated AML cells, in which actin cytoskeleton remodeling by FLT3-dependent RAC1 activation, N-WASP phosphorylation and ARP2/3 complex activation favors

AML cell survival and proliferation. Moreover, RAC1 hyperactivation inhibits apoptosis by affecting MCL1/BCL2 expression, changes the mechanical properties of these cells and increases the adhesion forces of *FLT3*-ITD AML cells to MSCs (Fig. 6a). Based on these findings, under the condition that this combination is safe and feasible, we see clinical potential in combining midostaurin, venetoclax and Eht1864 as a novel therapeutic strategy to treat FLT3-mutant AML patients (Fig. 6b).

**Fig. 3 FLT3 dependent RAC1 activation promotes dysregulation of actin polymerization regulators in Midostaurin-resistant cells. a** The protein expression analysis of actin polymerization positive (red) and negative (green) regulators by western blot in MV4-1/MOLM-13 MID-Res compared to parental cells, in MV4-11/MOLM-13 MID-Res ± 50 nM Mid compared to MID-Sens ± 50 nM Mid. **b** In FLT3 KD MV4-11/MOLM-13 MID-Res compared to scramble MV4-11 MID-Res and in MV4-11 MID-Res treated with Mid and Eht alone/combination. GAPDH was used as housekeeping. **c** Proliferation and Mid IC50 (after 4 days) in MV4-11/MOLM-13 MID-Res N-WASP KD ((MV) $p < 0.0001$, $t = 25.7$, df $= 3.8$, (MO) ($p < 0.001$, $t = 10.5$, df $= 3.9$)), WAVE2 KD ((MV) $p = 0.32$, $t = 1.1$, df $= 2.93$, (MO) $p = 0.4$, $t = 1$, df $= 2.6$), ARP2/3 complex KD ((MV) $p < 0.001$, $t = 39.6$, df $= 2.06$, (MO) $p < 0.001$, $t = 11.8$, df $= 3.6$), and PFN1 KD ((MV) $p < 0.001$, $t = 25.1$, df $= 3.9$, (MO) $p < 0.001$, $t = 9.8$, df $= 3.5$) compared to scramble MID-Res cells. **d** Actin polymerization positive regulators protein expression analysis targeting them by siRNAs in MV4-11/MOLM-13 MID-Res. **e** Correlation analysis between the mean of actin polymerization positive regulators (N-WASP, WAVE2, ARP2, PFN1) and FLT3 pathway genes expression in 639 AML patients by using microarray expression data (E-MTAB-3444). The western blot results are normalized by loading control (GAPDH) and are expressed as fold change relative to the control. Data are shown as means ± SDs (error bars) from three independent experiments.

## Material and methods

**Human samples and cell lines.** The human AML cell line MV4-11 (*homozygous FLT3-ITD*) and MOLM-13 (heterozygous *FLT3-ITD*) were used for in vitro studies. Midostaurin-resistant cell lines were developed as follows: (1) $1 \times 10^6$/ml MV4-11 and MOLM-13 cells were plated in 24-well plates in 2 ml cell culture medium and treated with 10 nM Midostaurin, (2) every 7 days, leukemia cells were adjusted to $1 \times 10^6$/ml in fresh medium and 2 ml of this cell suspension plated per well in 24-well plates, and (3) MV4-11 and MOLM-13 cells were initially exposed to 10 nM Midostaurin and concentration was increased serially (increased by 10 nM weekly) up to 50 nM. MV4-11 and MOLM-13 parental cell lines were purchased from ATCC and DSMZ, respectively. They were maintained in culture in RPMI-1640 medium supplemented with 10% fetal bovine serum with 1% penicillin-streptomicin and 1% L-Glutamine at 37 °C in a humid atmosphere containing 5% $CO_2$ at a concentration of $2 \times 10^5$ to $1 \times 10^6$. HS-5 MSC line was kindly provided by Patricia Maiso laboratory and were maintained in DMEM medium supplemented with 10% fetal bovine serum with 1% penicillin-streptomicin 1% L-Glutamine at 37 °C in a humid atmosphere containing 5% $CO_2$ at a concentration of $5 \times 10^5$ to $1 \times 10^6$. The cell lines were tested for mycoplasma contamination monthly. BM mononuclear cells were obtained from refractory FLT3-mutated AML patients after an informed consent was signed by the patients or the patient's guardians, in accordance with the Declaration of Helsinki. Clinical characteristics of relapsed FLT3-mutated AML patients used in this study are explained in Table S1. To test the toxicity of the compounds that were used in this work, we used PBMCs obtained from three healthy donors.

**dSTORM analysis of FLT3 receptor expression.** Localization data were analyzed by custom-made code written in Mathematica 11.1 (Wolfram Research Inc.). From each dSTORM image an appropriate region of interest was identified and localizations within this region of interest were selected. A DBSCAN clustering algorithm (Elki Data Mining) was applied to group localizations according to their local density[48]. Parameters were chosen by scanning the parameter space for weakest dependency of the resulting clustering set (neighborhood radius or epsilon = 20; minPoints = 3).

*Dye conjugation to antibodies.* After buffer exchange using 0.5 ml 7 kDa Spin Desalting Columns (Thermo Fisher, 89882, Erlangen, Germany) in 100 mM $NaHCO_3$ 50 μg of the purified monoclonal anti-FLT3 (BD Biosciences 558995, Heidelberg, Germany) were incubated in a five molar excess with Alexa Fluor 647 NHS (Thermo Fisher Scientific, A20101MP, Erlangen, Germany) at RT for 3 h in the dark. To exchange the buffer to 0.02 $NaN_3$ and to remove unreacted dyes the antibodies were purified with 0.5 ml 7 kDa Spin Desalting Columns. Lastly, the degree of labeling of the conjugated antibody was measured by a UV–vis

spectrophotometer (Jasco V-650). The conjugated antibodies were stored at 4 °C.

*Live cell immunostaining.* For super-resolved imaging $1.5 \times 10^5$ MV4-11 cells per well were seeded in poly-D-lysine coated chambers (Lab-Tek II, Nunc, Thermo Fisher Scientific, Erlangen, Germany) and the cells adhered at 37 °C at 5% $CO_2$ for 1 h. For live cell staining the cells were incubated for 30 min on ice in an antibody concentration of 2.5 μg/ml in PBS. Following by washing of the cells and fixation for 15 min in 2% formaldehyde and 0.2% glutaraldehyde. After three more washing steps the cells were stored at 4 °C until in PBS imaging.

*dSTORM imaging.* The dSTORM samples were imaged on a homebuilt widefield setup with an inverted microscope (Olympus IX-71) using an oil immersion objective (Olympus APON 60xO TIRF, NA 1.49). The dyes were excited with a laser of the wavelength 639 nm (Genesis MX639-1000, Coherent) The excitation light was filtered from the fluorescence light by a beam splitter (ZT405/514/635rpc, Chroma) and an emission filter (Brightline HC 679/41, Semrock). Imaging was carried out with an EMCCD camera (iXon Ultra 897, Andor) for 15,000 frames at a rate 50 Hz at ~7 kW/cm in 100 mM ß-mercaptoethylamin pH 7.4. The super-resolved dSTORM images were reconstructed with the open source software rapidSTORM 3.3[49].

**RAC1 activation assay.** RAC1 activation was measured by RAC1 G-LISA™ kit (Cytoskeleton BK126, Denver, CO, USA). This kit provides RAC1-GTP-binding protein linked to the wells of a 96 wells plate. The bound active RAC1 is detected with a RAC1 specific antibody and luminescence. The RAC1 activation was analyzed in MV4-11/MOLM-13 MID-Res ±50 nM Midostaurin compared to their parental cells ±50 Nm Midostaurin and FLT3 KD MV4-11/MOLM-13 MID-Res versus scramble MV4-11/MOLM-13 MID-Res. Following the kit instructions, these cells were incubated in serum free medium during 24 h to inactivate RAC1 In the case of FLT3 KD and scramble MV4-11/MOLM-13 MID-Res cells serum starvation was done 48 h after siRNAs transfection, to be sure that FLT3 was knocked down. Then, $5.10^6$ MID-Sens/MID-Res cells ±50 nM Midostaurin during 30 min (At this time were observed the highest differences). In FLT3 KD experiment, $5.10^6$ FLT3 KD/scramble MID-Res cells after serum starvation, were placed in supplemented RPMI-1640 medium at 37 °C in a humid atmosphere during 30 min. Then, the cells were harvested, lysates and protein was quantified by Precision Red™ Advanced Protein Assay Reagent (Cytoskeleon GL50). The final concentration used of equalized lysates was 1 mg/ml. Anti-RAC1 primary and the secondary HRP labeled antibody were diluted 1:5000. This antibody dilution allowed that readings were in the linear range of the luminometer. RAC1 activation Relative Light Units (RLU) was calculated by resting the lysis Buffer RLU mean (used as reference blank) to MID-

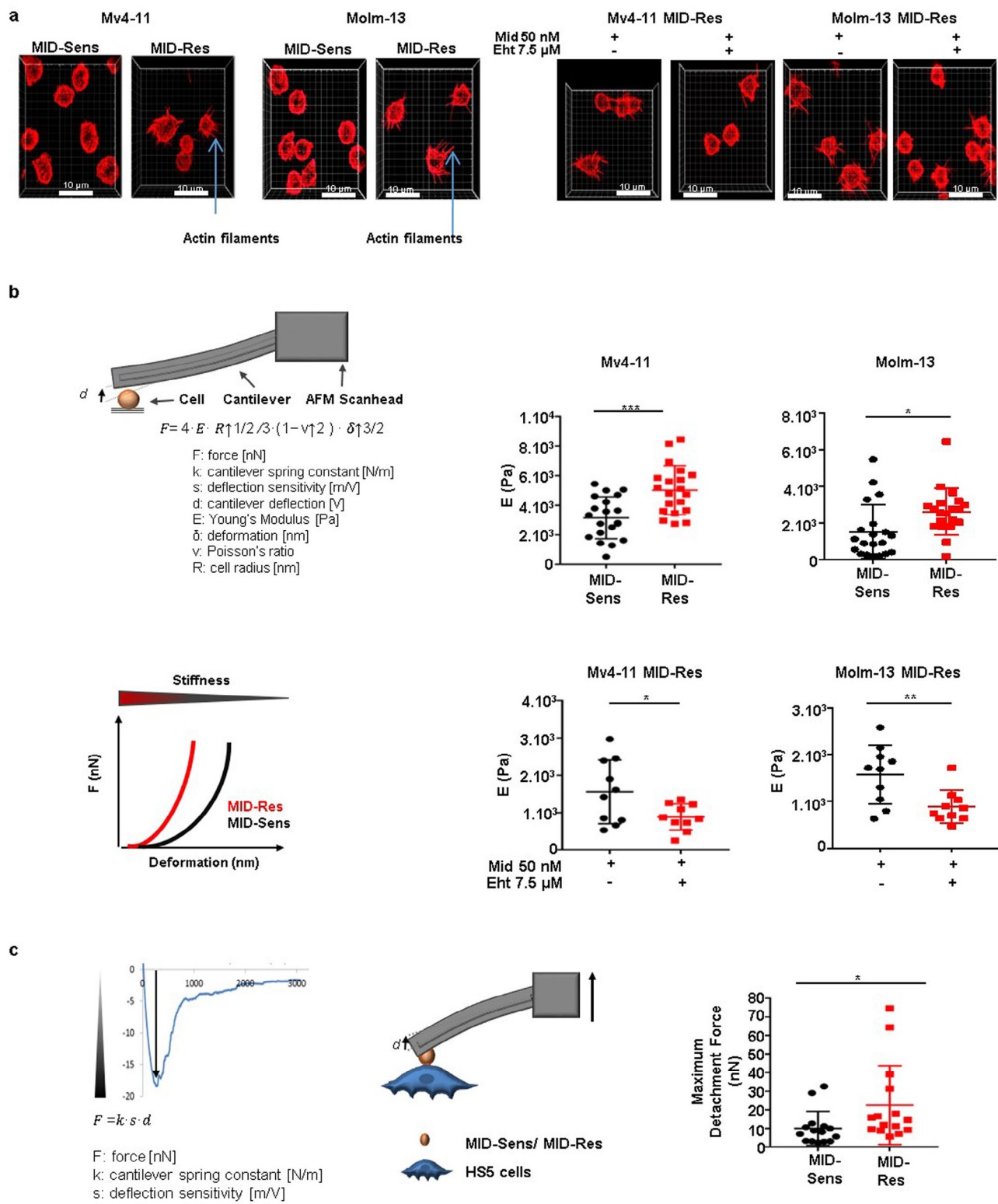

**Fig. 4 Deregulation of cell mechanical properties and enhancement of cell adhesion to mesenchymal stromal HS-5 cells in MV4-11/MOLM-13 Midostaurin-resistant cells compared to sensitive cells. a** Actin filaments visualization in MV4-11/MOLM-13 MID-Res versus MID-Sens cells and MV4-11/MOLM-13 MID-Res treated with the combination 50 nM Mid + 7.5 µM Eht versus MID-Res treated with 50 nM Mid + DMSO during 24 h. **b** Cell stiffness analysis in MV4-11/MOLM-13 MID-Res compared to MID-Sens cells ((MV) $p < 0.001$, $F = 14.8$, df = 1, (MO) $p < 0.05$, $F = 6.4$, df = 1) and in MV4-11/MOLM-13 MID-Res treated with the combination 50 nM Mid + 7.5 µM Eht versus MID-Res treated with 50 nM Mid + DMSO (vehicle control) during 24 h ((MV) $p < 0.05$, $F = 5.3$, df = 1, (MO) $p < 0.01$, $F = 9.3$, df = 1)*. **c** Cell adhesion forces quantification in MV4-11 MID-Sens/HS-5 cells and MV4-11 MID-Res/HS-5 ($p < 0.05$, $F = 4.3$, df = 1)*. Data are shown as means ± SDs (error bars) from three independent experiments. *Due to technical limitations, EHT1864 monotherapy was not performed in stiffness/cell adhesion assays.

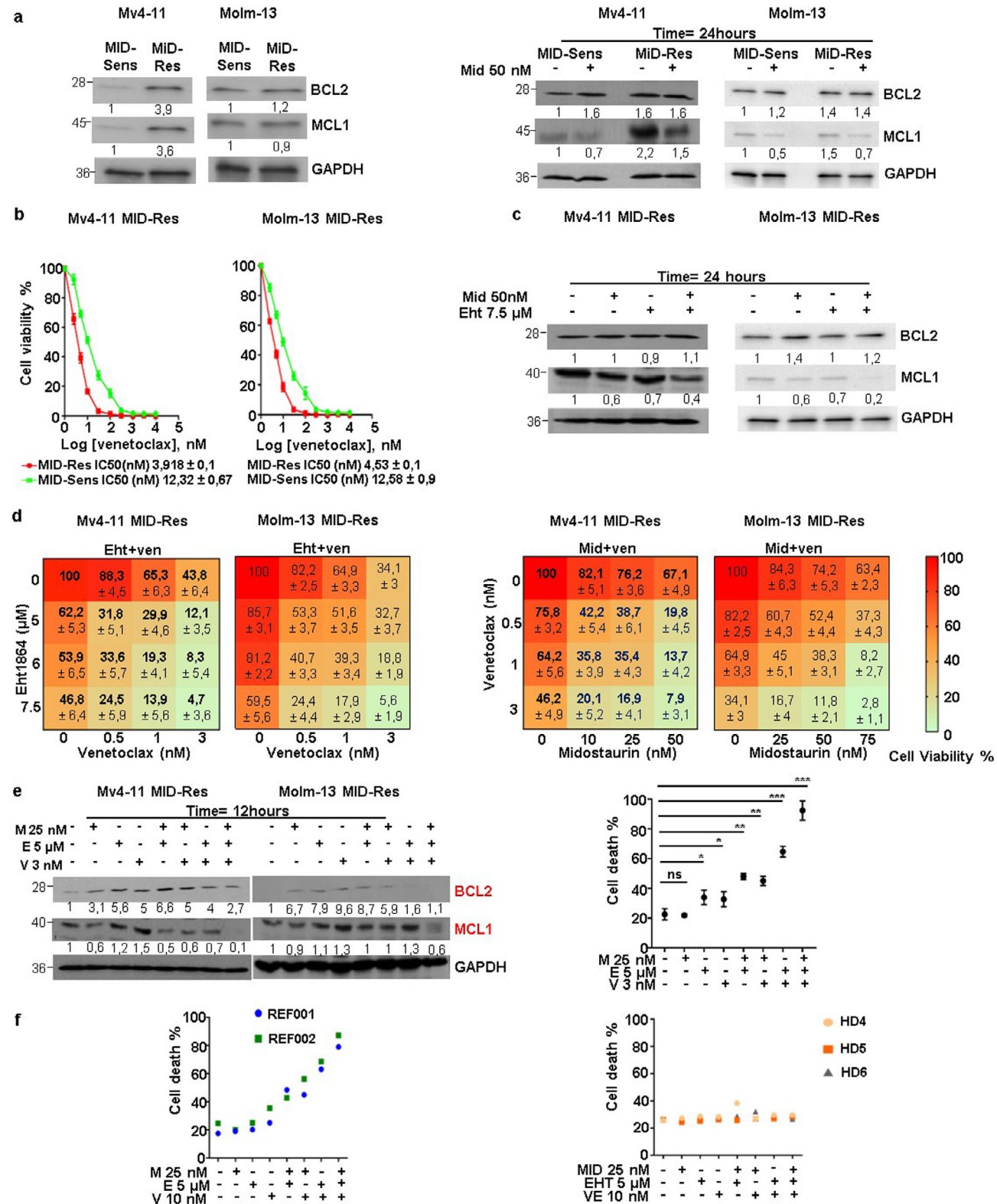

Sens/MID-Res ±Midostaurin RLU mean MID-Res scramble/MID-Res FLT3 KD RLU mean.

**Cell death and cell cycle analysis**. The cell death and cell cycle assay were carried out as described previously [50].

**Cell viability assay and cell treatments**. Cell viability was analyzed by CellTiter-Glo® Luminescent Cell Viability Assay (Promega,

Madison, WI, USA) following manufacturer's instruction. All experiments were repeated three times. ABT-199 (Venetoclax), EHT 1864 2HCl (Eht1864), PKC412 (Midostaurin) and CK-636 compounds were purchased from selleckhem (Munich, Germany). Cytochalasin D was purchased from Sigma-Aldrich (Munich, Germany). To understand the mechanism by which the combination of Midostaurin + Eht1864, Midostaurin + Venetoclax, and Eht1864 + Midostaurin + Venetoclax overcome Midostaurin resistance, MV4-11

**Fig. 5 The cell viability and MCL1/BCL-2 protein expression analysis in MV4-11/MOLM-13 Midostaurin-resistant cells treated with the triple combination Eht1864/Venetoclax/Midostaurin. a** BCL-2 and MCL1 protein expression analysis in MV4-11/MOLM-13 MID-Res compared to MID-Sens cells, in MV4-11/MOLM-13 MID-Res ± 50 nM Mid compared to MID-Sens ± 50 nM Mid. **b** Venetoclax (Ve) IC50 analysis in MV4-11/MOLM-13 MID-Sens and MID-Res cells treated with Ve during 48 h. **c** BCL-2 and MCL1 protein expression analysis in MV4-11/MOLM-13 MID-Res treated with Mid and Eht alone/combination. **d** Cell viability and synergy analysis in MV4-11/MOLM-13 MID-Res treated with Mid, Ve, and Eht alone/combination during 48 h. **e** BCL-2 and MCL1 protein expression and cell death analysis (Annexin V + /PI- + Annexin V + PI+) in DMSO (control), Mid ($p = 0.76$, $t = 0.34$, df = 2.3), Ve ($p = 0.054$, $t = 2.7$, df = 3.6), Eht ($p < 0.05$, 3.1, df = 3.7) and Mid + Eht ($p < 0.01$, $t = 10.29$, df = 3) Mid + Ve ($p < 0.01$, $t = 8$, df = 3.8) Ve + Eht ($p < 0.001$, $t = 14$, df = 3.9) and Eht + Ven + Mid ($p < 0.001$, $t = 16.1$, df = 3.2) during 24 h. **f** Cell death analysis by flow cytometry in REF001/REF002 AML samples and in PBMCs obtained from Healthy Donors HD4/HD5/HD6 treated with Mid, Eht, and Ve alone/combination during 24 h. The western blot results are normalized by loading control (GAPDH) and are expressed as fold change relative to the control. Data are shown as means ± SDs (error bars) from three independent experiments.

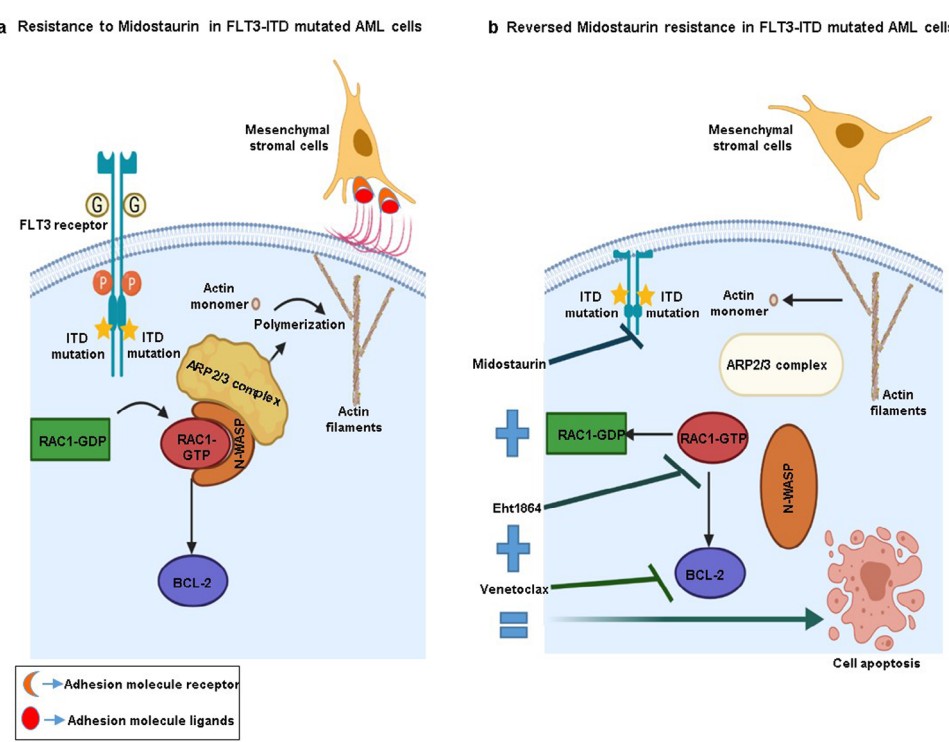

**Fig. 6 Summary results figure. a** Overactivation of actin polymerization machinery and antiapoptotic BCL-2 family in FLT3-ITD Midostaurin-resistant AML cells. **b** The triple combination Eht1864/Venetolax/Midostaurin by targeting Actin polymerization and BCL-2 reversed Midostaurin resistance.

and MOLM-13 MID-Res cells were treated at the half maximal inhibitory concentration (IC50) and maintained in culture for one day. Then, the cells were washed in PBS and used for different assays. The IC50 value was determined using GraphPad Prism software, log (inhibitor) versus response variable slope from dose-response inhibition (version 5, La Jolla, CA, USA). The synergistic between these compounds in Midostaurin-resistant cells was evaluated by Compusyn software.

**Correlation analysis between the expression of actin polymerization inducers and FLT3 signaling in AML patients.** Meta information and gene expression data of 639 AML patients were obtained from ArrayExpress (E-MTAB3444)[51]. Raw data were RMA normalized using the {affy} package and multiple probes per gene were aggregated by mean. FLT3 target gene expression was correlated with the expression of N-WASP, WASF2, PFN1, ARP2 or the mean expression of these four genes ("inducers of actin polymerization") dependent on the *FLT3* mutation status of the patient. FLT3 target genes ($n = 61$) were extracted from PathCards (https://pathcards.genecards.org/; using the "FLT3 signaling" SuperPath[52].

**Cell stiffness assay.** The stiffness of MID-Sens and MID-Res cells was assessed by using FluidFM® (Cytosurge AG, Glattbrugg (Zurich), Switzerland), a new technology adapted to traditional AFM in a Flex-FPM system (Nanosurf GmbH, Langen, Germany). It is mounted on an Axio Observer Z1 inverted microscope (Carl Zeiss, Germany) and a piezoelectric stage of 100 μm retraction range. Micropipette cantilevers (Cytosurge AG, Switzerland) of 8 μm aperture diameter and 0.33 N/m nominal spring constant were used. For the measurements of the stiffness of cells, a cell was immobilized at the aperture of the cantilever by applying a soft underpressure of 30 mbar, then, approaching at 500 nm/s, it was softly pressed against the culture dish until a set point of 3 nN was reached, while acquiring the data points at 3500 Hz. Then the cantilever was retracted, cleaned with NaClO 5% and thoroughly rinsed 4 times with ultrapure water and PBS. The procedure was repeated with 20 cells. The data points corresponding to the approach curves were extracted and processed in MATLAB 2017b (Mathworks, Natick, MA, USA) and fitted with the Hertz Model to obtain the modulus of elasticity, as indicator of the cell stiffness, as explained before[53]. The cell stiffness was measured in Midostaurin-resistant compared to

sensitive cells and resistant cells treated with the combination 7.5 μM Eht1864 + 50 nM Midostaurin versus resistant cells + 50 nM Midostaurin during 24 h.

**Adhesion forces assay**. In total, $1.10^6$ HS-5 cells were cultured in a 6 well plate for 24 h. Then, $5.10^5$ MID-Sens and MID-Res cells were independently co-cultured with the stromal layer in serum free conditions for 24 h. Afterwards, we quantified and compared the cell adhesion forces of MID-Res and MID-Sens onto HS-5 cells. The quantification of adhesion forces was performed by using the above-described FluidFM® system. Micropipette cantilevers of 4 μm aperture diameter and a nominal spring constant of 0.4 N/m were used. MV4-11 MID-Sens or MID-Res cells were approached with the cantilever at a speed of 5 μm/s until it pressed the surface of a cell with a set point of 3 nN. Next, the cell was immobilized at the cantilever aperture by applying an under pressure of 300 mbar, and retracted at 5 μm/s until complete detachment from the stromal layer. The cantilever was cleaned with NaClO 5% and thoroughly rinsed with water and PBS after measuring each cell. The data acquired during the retraction of the cantilever were extracted with SPIP 6.2.0 software (Image Metrology, Copenhagen, Denmark) and the cell-cell adhesion force was calculated according to Hook's Law, as explained before[53].

**siRNAs electroporation in AML cell lines**. SiRNAs against FLT3 receptor (Reference: HSS103748) and actin polymerization inducers (ARP2 (Reference: HSS115366), N-WASP (Reference: HSS113263), WAVE2 (Reference: HSS145484) and PFN1 (Reference: HSS107879) were purchased at Thermo Fischer Scientific (Erlangen, Germany). SiRNAs were dispensed into 1.5 ml Eppendorf tubes and mixed with the cell suspension by gentle pipetting. The complete range of electroporation mixes was prepared and electroporation carried out with a Gene Pulser (Bio-Rad Laboratories, Munich, Germany) at a capacity setting of 960 μF and with 300 voltage.

**Flow cytometric analysis of FLT3 receptor expression**. Cell surface expression of FLT3 (CD135) was analyzed using a PE-conjugated mouse anti-human FLT3 monoclonal antibody (clone 4G8, BD Biosciences, Heidelberg, Germany). In brief, $1 \times 10^6$ cells in 100 μl FACS Buffer (1X PBS + 1% FCS + 5 mM HEPES) were stained with 20 μl of PE-FLT3 (the optimum concentration was determined after titrating the antibody) for 30 min at 4 °C. After washing the cells, a viability dye (1:1000) was used (Cat. No. L34960, Thermo Fisher Scientific, Erlangen, Germany) for 30 min at room temperature. Cells were washed and resuspended in 300 μl FACS Buffer and analyzed using the CytoFLEX flow cytometer (Beckman Coulter). 50000 living cells were recorded. The gating strategy consisted of debris removal (FSC/SSC), double discrimination (FSC-A/H), dead cell exclusion (APC-) and FLT3+. Analysis was performed using FlowJo V.10.6 (BD Biosciences).

**Immunoblotting**. Cells were harvested and lysed using lysis buffer (Thermo Fischer Scientific), reconstituted, and whole-cell lysates were subjected to SDS-PAGE and transferred to Nitrocellulose Blotting Membrane (GEhealthcare Life science). Anti-Phospho Tyrosin 969 FLT3 (working dilution 1:1000), Anti-ARP2 (working dilution 1:1000), anti-N-WASP (working dilution 1:1000), anti-WAVE2 (working dilution 1:1000), anti-PFN1 (working dilution 1:1000), anti–MCL1 (working dilution 1:1000), anti–BCL-2 (working dilution 1:1000), and anti-GAPDH (working dilution 1:2000) antibodies were purchased from Cell Signaling Technology; The HRP-conjugated secondary antibody (working dilution 1:5000) were from Cell Signaling Technology.

Moreover, it was used anti-P-N-WASP [Ser484/Ser485] (working dilution 1:1000) and anti-P-WAVE2 [Ser343] (working dilution 1:1000) from Merck. The quantification of western blots were done by ImageJ software: (1) We measure the bands and their backgrounds along with the loading control bands (GAPDH) and their backgrounds, (2) Invert the pixel density for all data, the inverted value is expressed as 255—X, (3) after doing the inversions, it was calculated the net value of our net bands and GAPDH by substracting the inverted background, and (4) was measured fold change of net band/GAPDH relative to the control (scramble in siRNAs experiments and untreated sample in treatments experiment.

**Immunofluorescence of actin filaments**. In total, $1.5 \times 10^5$ cells were seeded on poly-D-lysine (Sigma, P6407-5MG, Munich, Germany) coated Lab-Tek chambers (Lab-Tek II, Nunc, Thermo Fisher Scientific, Erlangen, Germany) and adhered for 1 h at 37 °C and 5% $CO_2$. Cells were fixed using a cytoskeleton buffer (CB, 10 mM MES buffer pH 6.1, 150 mM NaCl, 5 mM EGTA, 5 mM glucose, as previously reported[54], with an optimized protocol designed for cytoskeleton staining. Here, the cells were incubated for 1 min in CB1 (CB + 0.25% Triton + 0.3% glutaraldehyde) followed by a 10 min incubation in CB2 (CB + 2% glutaraldehyde). Thereafter autofluorescence was quenched by an incubation of 7 min in 0.1% $NaBH_4$ and unspecific binding was blocked by a 30 min incubation in 5% BSA (Sigma, A3983-100g) after three washing steps. The cells were stained overnight at 4 °C in 0.1 μM Atto643-Phalloidin (ATTO-TEC, AD 643-81) in 5% BSA and stored at 4 °C. Prior to imaging the cells were washed again and imaged in PBS using a Zeiss Elyra S.1 SIM. The z-stackes were rendered and processed with Imaris 8.4.1.

**Statistics and reproducibility**. All the experiments were repeated three times (Biological triplicates). Data are expressed as means ± standard deviation. Statistical significance between two samples was estimated with Student's $t$ test (two-tailed). Differences with $p$ values $< 0.05$ were considered as significant ($*p < 0.05$, $**p < 0.01$, $***p < 0.001$, and $****p < 0.0001$). Statistical significance in stiffness and adhesion forces assays was estimated with one way ANOVA test. Statistical analyses were performed utilizing GraphPad Prism 7.0 (GraphPad Software, Inc., San Diego, CA).

**Reporting summary**. Further information on research design is available in the Nature Research Reporting Summary linked to this article.

## Data availability
The data that support the findings of this study are available within the manuscripts and its Supplementary files. Source data are provided in Supplementary Data 1. Meta information and gene expression data of 639 AML patients were obtained from ArrayExpress (E-MTAB3444) [51].

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

## Acknowledgements

This research was supported by the IZKF grant (B-334). As well, we thank Patricia Maiso Laboratory (Marta Lasa and Sarai Sarvide) for providing HS-5 cell line and the protocol for co-culturing mesenchymal stromal cells and tumor cells.

## Author contributions

Conception and design: A.G.T., A.S., J.G., H.E., S.K., and K.M.K. Development of methodology: A.G.T., A.S., R.G., P.E., S.W., H.J., and J.G.P. Acquisition of data (provided reagents, facilities, etc.): A.G.T., A.S., H.J., S.W., M.S., and J.G. Reagents and materials: E. T. and N.R. Analysis and interpretation of data (statistical analysis, biostatistics analysis): A.G.T., A.S., E.A., R.G., and P.E. Provided and managed patients: M.C.D., S.K., and R.T. Writing, review, and/or revision of the manuscript: A.G.T., A.S., R.G., P.E., L.H., A.R., M. S., L.R., M.H., J.G., H.E., S.K., and K.M.K. Study supervision: A.G.T., S.K., and K.M.K. All authors critically reviewed and approved this paper.

## Funding

## Competing interests

The authors declare no competing interests.
