## [Peer Review File · Communications Biology]

Reviewers' comments:

Reviewer #1 (Remarks to the Author):

In the present work, Garitano-Trojaola et al focus on the study of Midostaurin resistance in FLT3-ITD+ AML. They show the impact of RAC1 activity and the derived cytoskeleton remodeling on the acquisition of resistance. More importantly, they show how such resistance can be reverted with the use of BCL2 and RAC1 inhibitors. Although the experiments are well thought and executed, there are some points that need to be addressed.

Major comments:

1. The role of FLT3 receptor glycosylation seems confusing in the text. It is demonstrated that resistant cell lines have increased glycosylation, but it is also stated that treatment with midostaurin increased this modification. Could the authors please clarify?
2. Images of FLT3 in figure 1B are very blurry and hard to see.
3. It is demonstrated that midostaurin does not change the phosphorylation or activity of FLT3 receptor in resistant cell lines. Nevertheless, it is not clear in the manuscript what it has biological effects on its own: while in figure 2B (left) it seems to induce caspase 3, in the right part of the figure, no effect on the percentage of apoptotic cells is indicated for the same cell line (MV4-11 MID-Res). Furthermore, u
4. Does figure 2C depict caspase 3 or cleaved caspase3?
5. It is shown how cell stiffness and adhesion decreases with midostaurin and Eht1864 treatment compared to treatment with midostaurin alone. What was the effect of Eht1864 treatment alone? It should also be shown in figure 4.
6. In the manuscript, it is shown that BCL-2 expression, which is positive regulated by RAC1, is increased by midostaurin treatment in sensitive cell lines. This information is confusing, as treatment of these cells with the FLT3 receptor inhibitor decreases FLT3 activity and RAC1 activation. Please, clarify these results.

Minor comments:

1. The legend in figure S1 is missing. Please include it.
2. The line colors in figure 1A seem reversed, as the sensitive cell lines seem more resistant in the graph. Please double check.

Reviewer #2 (Remarks to the Author):

The manuscript, "Actin cytoskeleton deregulation confers midostaurin resistance in FLT3 mutant Acute Myeloid Leukemia," by Garitano-Trojaola et al., is an interesting investigation of a mechanism of midostaurin resistance associated with RAC1-dependent actin cytoskeleton remodeling.

I have the following suggestions/concerns:

1. The involvement of RAC1 in resistance to the FLT3 inhibitor, midostaurin, is indeed interesting and could have significant implications for clinical development of combination strategies geared toward overriding FLT3 inhibitor resistance. Wu et al. demonstrated (Haematologica 2019), FLT3-ITD cooperates with Rac1 to modulate the sensitivity of leukemic cells to chemotherapeutic agents via regulation of DNA repair pathways. Did authors explore the contribution of DNA repair molecules to midostaurin resistance, or only factors associated with actin polymerization? Would authors predict actin cytoskeleton deregulation to generalize to resistance conferred by other FLT3 inhibitors, namely more targeted agents such as quizartinib, crenolanib or gilteritinib? Since gilteritinib is FDA-approved, this information may be useful for clinicians.
2. The first paragraph of the introduction could benefit from more details about FLT3 inhibitors. Gilteritinib, which has been approved, is not mentioned. What do authors mean by "secondary

tyrosine kinase inhibitors"? There are first and second generation FLT3 inhibitors and the ones authors listed represent both.

3. Page 9, "The FLT3 KD reduced significantly the RAC1 activation in MV4-11 and MOLM-13 MID-Res cells (Fig. 1F)." KD efficiency is typically shown by western analysis.

4. Figure 1B. FLT3-ITD signals are weak and diffuse, and it is hard to see the doublet.

5. Figure 1C. There is no increase in FLT3 expression in Res versus sensitive cells. There is only an increase in FLT3 expression in the presence of 50 nM midostaurin. The text says "As expected, FLT3 surface expression measured by flow cytometry was increased in MV4-11 and MOLM13 MID-Res cells versus parental cells." If authors are referring only to the drug-treated cells they should state this more clearly.

6. Page 11: "MV4-11/MOLM-13 MID-Res N-WASP and ARP2 KD compared to MID-Res scramble increased the expression of CFL1 phosphorylation (Fig. 3D)." This does not appear to be true for MOLM13 MID-RES cells. It is only true for MV4-11 MID-RES cells.

7. Figure 5B. The quality of the signals for Bcl2 and MCL1 in MOLM13 is relatively poor. Any drug effects authors are describing are small. Densitometry would be beneficial.

8. Figure 3: A blot showing FLT3 KD efficiency is needed.

9. Figure 3. Some of the protein changes shown are small. Authors should do densitometry.

10. Figure 5A. "BCL-2 and MCL1 were strongly upregulated in MV4-11/MOLM13 MID-Res compared to their parental cells (Fig. 5A)." The data are not consistent between MV4-11 and MOLM13. There is a strong upregulation of Bcl-2 and Mcl1 in resistant MV4-11 but not in MOLM13. Authors should describe their data in such a way as to avoid over-generalization. There is no change in Mcl1 or Bcl2 in Res MOLM-13 versus sensitive MOLM13 in Figure 1A.

11. Figure 5C. The Mcl1 signal in Res MOLM13 cells is very light and it is difficult to see drug combination effects.

12. Figure 5E. GAPDH levels are lower in the triple drug administration lanes, suggesting drug toxicity/nonspecific effects. Authors should consider putting the healthy donor PBMC data into the main figure to strengthen their argument that the combination is not toxic.

13. Page 17. "Notably, no apoptosis was observed in PBMCs obtained from healthy donors treated with the triple combination." If this is in the discussion, the data should be in the main manuscript.

14. Page 18. "Based on these findings, we see clinical potential in a triple combination, Midostaurin, Venetoclax and Eht1864 as a novel therapeutic strategy to treat FLT3 mutant AML patients." If this is a main conclusion of this study, convincing data suggesting non-toxicity of the triple combination treatment should be included in the main manuscript.

15. There is no Figure 2F shown in Figure 2 or the figure legend.

16. Page 11, line 273: "The IC50 of midostaurin was significantly reduced in MV4-11/MOLM-13 MID-Res (Fig. 3C, Fig S2A)."

17. Page 3, line 77: Need to introduce the name in full of RAC1 and the abbreviation sooner. Ras-related C3 botulinum toxin substrate 1 (RAC1) should go earlier in the line (before "RAC1 activation").

18. Page 3, line 70: tyrosine is spelled wrong (missing an "e") This also occurs in a few other places in the manuscript (Page 9, line 213).

19. Page 3, line 84: "complex that includes the RAC1+P-WAVE2 or the P-NWASP..."

20. Page 4, lines 90-93: Grammar needs to be improved.

21. Page 4, line 95: "... associated with FLT3 inhibitor resistance, may induce RAC1 hyper-activation..."

22. Page 8, line 195: "Statistics"

23. Page 8, lines 204-206. Need hyphens. "FLT3-dependent RAC1 hyper-activation in FLT3-ITD+ Midostaurin-resistant cells." "In order to study the role of RAC1-regulated cellular processes..." "...we developed Midostaurin-resistant AML cell lines from..."

Reviewer #3 (Remarks to the Author):

In this study, Garitano-Trojaola A et al showed that RAC1 hyperactivation leads to resistance to midostaurin, a first generation FLT3 inhibitor, through hyper-/hypo-phosphorylation of actin polymerization positive/negative regulators N-WASP/CFL1 and anti-apoptotic BCL-2. The authors then showed that targeting of actin polymerization regulators could overcome resistance to midostaurin. The concept of the study is interesting. However, there are numerous technical and data presentation issues that must be addressed before it can be considered for publication in the journal. Specific comments are listed below.

Major points:

1. Introduction, lines 70-73: Please explain the meaning of secondary tyrosine kinase inhibitors. In addition, the second generation FLT3 inhibitor, Gilteritinib, has been approved by the US FDA to treat refractory and relapsed FLT3-mutated AML. Why it is left out in the introduction?
2. Materials and Methods, lines 107-111: Please provide the correct source of MOLM-13 cell line. ATCC does not have this cell line. Further, please state if these cell lines are tested for mycoplasma contamination on a regular basis.
3. Results, lines 209-210: How did you determine midostaurin IC50s in these cells? There is no description in the Materials and Methods section and the legend of Figure 1. I strongly recommend the authors to move the description of cell viability assay shown in the supplement to the main text.
4. Figure 1A: The IC50 values shown below the graphs make no sense. Based on these numbers, there is no way the fold-difference can be 366 and 298.
5. Figure 1B: The quality of the blots is too poor to draw any meaningful conclusion.
6. Figure 1C: The data is presented in a very confusing way. It is hard to tell which cells were treated with midostaurin.
7. Figure S1: The legend is too brief to tell how the experiments were performed. Did the authors treat the cells with midostaurin after FLT3 KD in the MID-Res cells? If not, what do you mean by FLT3 KD reversed midostaurin resistance?
8. Figure 2A: The data shown in this panel is not consistent with the text in the main manuscript. The graph and legend show that cell viability was measured, while the text in the main manuscript says cell death was measured. In addition, MV4-11 and MOLM-13 should be changed to MV4-11 MID-Res and MOLM-13 MID-Res, respectively.
9. Figure 2C: How did the authors define apoptotic cells? Only annexin V+/PI- negative cells are true apoptotic cells. Further, the baseline values of apoptotic cells are very high (20-27%), indicating that the cells were not healthy when the experiments were performed. Furthermore, did the two drugs synergize in inducing cell death?
10. Results, lines 246-247: Cell viability assay only measures number of viable cells and cannot determine cell death. Decrease of viable cells can be attributed to cell growth inhibition and/or cell death.
11. Figure 2D&E: If the data shown in the left panels of panels 2D and 2E were from the same

primary patient samples, the cell viability data and cell apoptosis data don't match. How could a cell population with 52.7% viable cells is 70% apoptotic?

12. Figure 2E: Were these primary patient samples? What do the symbols represent?

13. Results, line 251: This subtitle indicates that FLT3-ITD+ cells and FLT3-wt cells were compared.

14. Figure 3: Densitometry for the blots should be perform.

15. Results, lines 324-325: The authors state "BCL-2 and MCL1 were strongly up regulated in MV4-11/MOLM-13 MID-Res compared to their parental cells (Fig. 5A)". The data in MV4-11 cells supports this statement, while the data in MOLM-13 cells does not. Further, the quality of some the blots shown in Figures 5A and 5C is too poor. Therefore, no meaningful conclusions can be draw based on these blots.

16. There is no animal study data to support that targeting RAC1 is tolerable, especially in combination with midostaurin and venetoclax, and can overcome midostaurin resistance in vivo.

17. The manuscript will benefit from professional English editing.

Minor points:

1. Abstract, lines 44-45: please add the word "to" after leads.

2. Introduction, line 77: The abbreviation RAC1 should not be used before the full name of the protein appears in the text.

Reviewer #1 (Remarks to the Author):

In the present work, Garitano-Trojaola et al focus on the study of Midostaurin resistance in FLT3-ITD+ AML. They show the impact of RAC1 activity and the derived cytoskeleton remodeling on the acquisition of resistance. More importantly, they show how such resistance can be reverted with the use of BCL2 and RAC1 inhibitors. Although the experiments are well thought and executed, there are some points that need to be addressed.

Major comments:

1. The role of FLT3 receptor glycosylation seems confusing in the text. It is demonstrated that resistant cell lines have increased glycosylation, but it is also stated that treatment with midostaurin increased this modification. Could the authors please clarify?

According to our and published data, FLT3 receptor glycosylation is induced by FLT3 inhibitors therapies (PMID: 28895560, 22927251 and 29472720), and increased glycosylation may contribute to FLT3 inhibitor resistance. This was described by Weisberg E. et al (reference 6 in our manuscript), who demonstrated FLT3 mutant Midostaurin resistant AML cells to show higher FLT3 receptor glycosylation, than sensitive cells (Figure 1B). Similarly, we observed increased FLT3 receptor glycosylation in our MID-Res cells. Moreover, inhibition of FLT3 glycosylation was associated with longer survival in mice with FLT3 mutant Leukemia (PMID: 22927251). We feel confident that the glycosylation of the FLT3 receptor plays an important role in the development of FLT3 inhibitor resistance, likely induced by therapy.

We thank the reviewer to rise this important question and we modified the text to clarify on this. As well, it was added a new reference (Number 40). Changes to the text are marked with yellow, lines 409-410.

2. Images of FLT3 in figure 1B are very blurry and hard to see.

We apologize for the insufficient figure 1B image quality and replaced it by an increased resolution western blot that includes quantification by ImageJ. The procedure of quantification is explained in the Supplemental Material&Methods, Immunoblot part.

3. It is demonstrated that midostaurin does not change the phosphorylation or activity of FLT3 receptor in resistant cell lines. Nevertheless, it is not clear in the manuscript what it has biological effects on its own: while in figure 2B (left) it seems to induce caspase 3, in the right part of the figure, no effect on the percentage of apoptotic cells is indicated for the same cell line (MV4-11 MID-Res).

We thank reviewer for this observation. Recently, James E Vince et al. have shown that MCL-1 inhibition through BAX/BAK activates caspase-3 (PMID: 30485804). Our data show that Midostaurin inhibits MCL-1 (Figure 1A), this could explain why midostaurin alone activates caspase 3 in MV4-11 MID-Res cells. Moreover, we think that BCL2 up-

regulation could be responsible for no apoptosis observed by Annexin V/PI. Additional experiments would be necessary to confirm this hypothesis.

4. Does figure 2C depict caspase 3 or cleaved caspase3?

Figure 2C shows a cleaved Caspase-3. This information was added to the figure legend. Changes to legend are marked with bold.

5. It is shown how cell stiffness and adhesion decreases with midostaurin and Eht1864 treatment compared to treatment with midostaurin alone. What was the effect of Eht1864 treatment alone? It should also be shown in figure 4.

We fully agree with the reviewer's point, however, the applied FluidFM® Cell Stiffness Assays and Adhesion Force Assays are truly "single-cell" techniques that, regrettably, do not allow to compare more than two conditions per experiment. Ideally, the conditions that should be included in this experiment would be 1) control cells (Midostaurin resistant cells), 2) MID-Res cells + Midostaurin, 3) MID-Res cells + EHT1864, and 4) MID-Res under EHT1864 and Midostaurin combination. However, as detailed in the procedure for the Cell Stiffness Assay in the original Supplementary Information, the drug is added to the samples, 24 hours before the measurements start. At this point, the measurements are launched having a maximal time window of 2 hours for the measurements to be done, otherwise the effect of the drug would be too strong and the measurements would not be comparable with each other. As we had to select the most relevant conditions, we chose as the first condition MID-Res cells under Midostaurin treatment (to make sure that cells were resistant to this compound) and, as the second condition, the combination that reverse Midostaurin resistance, EHT1864 together with Midostaurin.

In order to make this limitation of our study more visible to the reader we have moved the description of the protocol from the Supplementary Information into the Materials and Methods section in the main manuscript and added a cautionary note to the discussion. Changes to the text are marked with yellow, lines 210-227.

6. In the manuscript, it is shown that BCL-2 expression, which is positive regulated by RAC1, is increased by midostaurin treatment in sensitive cell lines. This information is confusing, as treatment of these cells with the FLT3 receptor inhibitor decreases FLT3 activity and RAC1 activation. Please, clarify these results.

We thank the reviewer for this helpful comment: RAC1 inhibits apoptosis through the binding and stabilizing of the anti-apoptotic BCL-2 family proteins, BCL-2, and MCL1 (17-19 references in the article). Our study shows that Midostaurin has the opposite effect on BCL-2 and MCL1 protein expression in MV4-11 parental cells: MCL-1 is decreased and BCL-2 is up-regulated.

This suggests that the reduction of RAC1 activation under Midostaurin treatment is essential on MCL1 but not on BCL-2 in MV4-11 parental cells. BCL-2 stability might

associate with other mechanisms. Further investigation is needed to elucidate the role of RAC1 in the stability of BCL2/MCL-1 in MID-Sens/MID-Res cells.

We thank the reviewer to rise this important question and we modified the text to clarify on this, lines 471-474.

Minor comments:

1. The legend in figure S1 is missing. Please include it.

We apologize and added the figure legend.

2. The line colors in figure 1A seem reversed, as the sensitive cell lines seem more resistant in the graph. Please double check.

We apologize and edited the figure line colors according to the reviewer's suggestion.

Reviewer #2 (Remarks to the Author):

The manuscript, "Actin cytoskeleton deregulation confers midostaurin resistance in FLT3 mutant Acute Myeloid Leukemia," by Garitano-Trojaola et al., is an interesting investigation of a mechanism of midostaurin resistance associated with RAC1-dependent actin cytoskeleton remodeling.

I have the following suggestions/concerns:

1. The involvement of RAC1 in resistance to the FLT3 inhibitor, midostaurin, is indeed interesting and could have significant implications for clinical development of combination strategies geared toward overriding FLT3 inhibitor resistance. Wu et al. demonstrated (Haematologica 2019), FLT3-ITD cooperates with Rac1 to modulate the sensitivity of leukemic cells to chemotherapeutic agents via regulation of DNA repair pathways. Did authors explore the contribution of DNA repair molecules to midostaurin resistance, or only factors associated with actin polymerization? Would authors predict actin cytoskeleton deregulation to generalize to resistance conferred by other FLT3 inhibitors, namely more targeted agents such as quizartinib, crenolanib or gilteritinib? Since gilteritinib is FDA-approved, this information may be useful for clinicians.

We thank the reviewer for this valuable comment.

Although, we did not explore the contribution of DNA repair molecules, different articles (PMID: 16767080, PMID: 28805828) have shown the importance of actin filaments in DNA repair genes, suggesting that DNA damage response inhibitors could increase FLT3 and actin cytoskeleton inhibitors response in FLT3 mutant AML. The reviewer's comment opens an interesting research line in AML. However, we focused our study on the role of the actin cytoskeleton in the resistance to Midostaurin in FLT3 mutant AML.

Regarding if Actin cytoskeleton deregulation could be resistant mechanism to all FLT3 inhibitors, we are currently developing Quizartinib, Crenolanib and gilteritinib resistant cell lines to answer the valuable question raised by the reviewer.

2. The first paragraph of the introduction could benefit from more details about FLT3 inhibitors. Gilteritinib, which has been approved, is not mentioned. What do authors mean by “secondary tyrosine kinase inhibitors”? There are first and second generation FLT3 inhibitors and the ones authors listed represent both.

We apologize for this inaccuracy and have modified our introduction according to the reviewers suggestion. Changes to the text are marked with yellow, lines 68-81 .

3. Page 9, “The FLT3 KD reduced significantly the RAC1 activation in MV4-11 and MOLM-13 MID-Res cells (Fig. 1F).” KD efficiency is typically shown by western analysis.

We thank the reviewer for this valuable comment. We have included respective western blots and were quantified by ImageJ software. The quantification procedure has been explained in Supplemental Material&Methods, Immunoblot part. The quantification results have been included in the blots figures.

4. Figure 1B. FLT3-ITD signals are weak and diffuse, and it is hard to see the doublet.

We apologize for the reduced figure 1B image quality and replaced it by an increased resolution western blot that includes quantification by ImageJ.

5. Figure 1C. There is no increase in FLT3 expression in Res versus sensitive cells. There is only an increase in FLT3 expression in the presence of 50 nM midostaurin. The text says “As expected, FLT3 surface expression measured by flow cytometry was increased in MV4-11 and MOLM13 MID-Res cells versus parental cells.” If authors are referring only to the drug-treated cells they should state this more clearly.

We agree with the reviewer and corrected the format of figure 1C. FLT3 is up-regulated in MID-RES vs MID-Sens in MV4-11 and MOLM-13 cell lines without Midostaurin induction.

6. Page 11: “MV4-11/MOLM-13 MID-Res N-WASP and ARP2 KD compared to MID-Res scramble increased the expression of CFL1 phosphorylation (Fig. 3D).” This does not appear to be true for MOLM13 MID-RES cells. It is only true for MV4-11 MID-RES cells.

We quantified the western blot from figure 3D by ImageJ and also CFL1 phosphorylation is increased in MOLM-13 MID-Res N-WASP (fold change relative to scramble= 1,8) and ARP2 (fold change relative to scramble= 1,5) KD.

7. Figure 5B. The quality of the signals for Bcl2 and MCL1 in MOLM13 is relatively poor. Any drug effects authors are describing are small. Densitometry would be beneficial.

We apologize for the insufficient image quality. We have redone the figures applying higher resolution.

8. Figure 3: A blot showing FLT3 KD efficiency is needed.

We thank the reviewer for this helpful suggestion, we have amended the blot to the figure.

9. Figure 3. Some of the protein changes shown are small. Authors should do densitometry.

According to the reviewer's suggestion we added the missing information. Western blot were quantified by ImageJ software.

10. Figure 5A. “BCL-2 and MCL1 were strongly upregulated in MV4-11/MOLM13 MID-Res compared to their parental cells (Fig. 5A).” The data are not consistent between MV4-11 and MOLM13. There is a strong upregulation of Bcl-2 and Mcl1 in resistant MV4-11 but not in MOLM13. Authors should describe their data in such as way as to avoid over-generalization. There is no change in Mcl1 or Bcl2 in Res MOLM-13 versus sensitive MOLM13 in Figure 1A.

We thank the reviewer for this cautionary note and edited our manuscript accordingly. Changes to the text are marked with yellow, lines 377-379.

11. Figure 5C. The Mcl1 signal in Res MOLM13 cells is very light and it is difficult to see drug combination effects.

We have included a new western blot and was quantified by ImageJ. Now, it is clearer observed MCL1 changes under EHT1864 alone/ combination Midostaurin + EHT1864 treatments.

12. Figure 5E. GAPDH levels are lower in the triple drug administration lanes, suggesting drug toxicity/nonspecific effects. Authors should consider putting the healthy donor PBMC data into the main figure to strengthen their argument that the combination is not toxic.

We included healthy donor PBMCs data under triple combination treatment in the main figures (Figure 5E). We appreciate the reviewer's comment, that helped that the current figure 5E further inforces our argument.

13. Page 17. “Notably, no apoptosis was observed in PBMCs obtained from healthy donors treated with the triple combination.” If this is in the discussion, the data should be in the main manuscript.

We agree with the reviewer's comment and we changed the manucsript accordingly (Figure 5E).

14. Page 18. “Based on these findings, we see clinical potential in a triple combination, Midostaurin, Venetoclax and Eht1864 as a novel therapeutic strategy to treat FLT3 mutant AML patients.” If this is a main conclusion of this study, convincing data suggesting non-toxicity of the triple combination treatment should be included in the main manuscript.

We fully agree with the reviewer, that safety is as important as efficacy when it comes to the clinical application. We therefore chose a more careful summarizing statement and thank the reviewer for this very helpful comment, line 498.

15. There is no Figure 2F shown in Figure 2 or the figure legend.

We apologize for this error and corrected our manuscript.

16. Page 11, line 273: “The IC50 of midostaurin was significantly reduced in MV4-11/MOLM-13 MID-Res (Fig. 3C, Fig S2A).”

We would kindly ask the reviewer to provide additional information to allow us to answer the reviewer’s question.

17. Page 3, line 77: Need to introduce the name in full of RAC1 and the abbreviation sooner. Ras-related C3 botulinum toxin substrate 1 (RAC1) should go earlier in the line (before “RAC1 activation”).

We appreciate your comment and edited the manuscript accordingly, line 86.

18. Page 3, line 70: tyrosine is spelled wrong (missing an “e”) This also occurs in a few other places in the manuscript (Page 9, line 213).

We apologize for this typo, that we corrected in the manuscript.

19. Page 3, line 84: “complex that includes the RAC1+P-WAVE2 or the P-NWASP...”

We apologize for this typo, that we corrected in the manuscript, lines 93-94.

20. Page 4, lines 90-93: Grammar needs to be improved.

21. Page 4, line 95: “... associated with FLT3 inhibitor resistance, may induce RAC1 hyper-activation...”

22. Page 8, line 195: “Statistics”

23. Page 8, lines 204-206. Need hyphens. “FLT3-dependent RAC1 hyper-activation in FLT3-ITD+ Midostaurin-resistant cells.” “In order to study the role of RAC1-regulated cellular processes...” “...we developed Midostaurin-resistant AML cell lines from...”

We apologize for the grammar errors and the typos referred in point 19-23. We corrected the manuscript and, in addition, sent the manuscript for professional language editing to improve readability.

Reviewer #3 (Remarks to the Author):

In this study, Garitano-Trojaola A et al showed that RAC1 hyperactivation leads to resistance to midostaurin, a first generation FLT3 inhibitor, through hyper-/hypo-phosphorylation of actin polymerization positive/negative regulators N-WASP/CFL1 and anti-apoptotic BCL-2. The authors then showed that targeting of actin polymerization regulators could overcome resistance to midostaurin. The concept of the study is interesting. However, there are numerous technical and data presentation issues that must be addressed before it can be considered for publication in the journal. Specific comments are listed below.

Major points:

1. Introduction, lines 70-73: Please explain the meaning of secondary tyrosine kinase inhibitors. In addition, the second generation FLT3 inhibitor, Giteritinib, has been approved by the US FDA to treat refractory and relapsed FLT3-mutated AML. Why it is left out in the introduction?

We agree with the reviewer, that Giteritinib has to be added to the introduction and have amended the introduction according to the reviewer's suggestion. Changes to the text are marked with yellow, lines 68-81.

2. Materials and Methods, lines 107-111: Please provide the correct source of MOLM-13 cell line. ATCC does not have this cell line. Further, please state if these cell lines are tested for mycoplasma contamination on a regular basis.

We thank the reviewer for this valuable comment. We amended the MOLM-13 source to the text. All cell lines were tested for mycoplasma contamination monthly. This information is given in material & methods, human samples and cell lines, lines 115 and 121-122.

3. Results, lines 209-210: How did you determine midostaurin IC50s in these cells? There is no description in the Materials and Methods section and the legend of Figure 1. I strongly recommend the authors to move the description of cell viability assay shown in the supplement to the main text.

Thank you for this important comment. The IC50 value was analyzed using GraphPad Prism software, log (inhibitor) versus response variable slope from dose-response inhibition (version 5, La Jolla, CA, USA). A detailed description of the cell viability assay and cell treatment part was moved to the main text, lines 187-199.

4. Figure 1A: The IC50 values shown below the graphs make no sense. Based on these numbers, there is no way the fold-difference can be 366 and 298.

We very much apologize for this error and corrected our manuscript accordingly. “The IC50 of midostaurin in MV4-11 cells increased from 15.09 nM to 55.24 nM in MID-Sens and MID-Res cells; that of MOLM-13 cells increased from 29.41 nM to 87.83 nM (Fig. 1A)”, lines 254-256.

5. Figure 1B: The quality of the blots is too poor to draw any meaningful conclusion.

We improved the image quality applying higher resolution.

6. Figure 1C: The data is presented in a very confusing way. It is hard to tell which cells were treated with midostaurin.

We apologize for being misinterpretable and we thank the reviewer for this helpful comment to improve our manuscript. To improve readability we adjusted the format of figure 1C. In this one can see that FLT3 is up-regulated in MID-RES vs MID-Sens in MV4-11 and MOLM-13 cell lines without Midostaurin induction.

7. Figure S1: The legend is too brief to tell how the experiments were performed. Did the authors treat the cells with midostaurin after FLT3 KD in the MID-Res cells? If not, what do you mean by FLT3 KD reversed midostaurin resistance?

We agree and expanded our figure legend.

8. Figure 2A: The data shown in this panel is not consistent with the text in the main manuscript. The graph and legend show that cell viability was measured, while the text in the main manuscript says cell death was measured. In addition, MV4-11 and MOLM-13 should be changed to MV4-11 MID-Res and MOLM-13 MID-Res, respectively.

We thank the reviewer for this comment, we edited the manuscript accordingly, line 289.

9. Figure 2C: How did the authors define apoptotic cells? Only annexin V+/PI-negative cells are true apoptotic cells. Further, the baseline values of apoptotic cells are very high (20-27%), indicating that the cells were not healthy when the experiments were performed. Furthermore, did the two drugs synergize in inducing cell death?

The reviewers point is well taken. To acknowledge the discussion whether annexin V+/PI+ positive cells should be late apoptotic cells or necrotic cells we edited the main figures and supplementary figures “apoptotic cells %” by “cell death %”. As well, in the legends, we have mentioned that cell death % was defined by Annexin V+/PI- + Annexin V+/PI+ cells.

To address the second, valuable reviewer’s point we confirm that, before we start any experiment, we measure the cell viability of our cell lines by Trypan Blue= Number of

live cells/ total number cells (Live cells+ death cells). The cell viability threshold of our cell lines was > 90 % to continue with the experiments. Moreover, caspase-3 cleavage assay showed no apoptosis in MV4-11 MID-Res cells.

Below attached synergy results, considering cell death. Also, Midostaurin and EHT1864 synergize in inducing cell death.

MV4-11 MID-Res; Combination Midostaurin+Eht1864 ; cell dead			
Dose MIDOSTAURIN	Dose EHT1864	Effect	CI
10.0	5.0	0.427	0.82313
10.0	6.0	0.573	0.79297
10.0	7.5	0.729	0.76451
25.0	5.0	0.587	0.64678
25.0	6.0	0.596	0.76541
25.0	7.5	0.701	0.80479
50.0	5.0	0.635	0.59975
50.0	6.0	0.755	0.58135
50.0	7.5	0.755	0.72668

Molm-13 MID-Res; Combination Midostaurin+EH1864; cell dead			
Dose MID	Dose EHT	Effect	CI
25.0	5.0	0.459	0.48906
25.0	6.0	0.434	0.64089
25.0	7.5	0.686	0.33017
50.0	5.0	0.53	0.38408
50.0	6.0	0.633	0.32224
50.0	7.5	0.833	0.16487
75.0	5.0	0.558	0.34921
75.0	6.0	0.729	0.22172
75.0	7.5	0.982	0.02204

10. Results, lines 246-247: Cell viability assay only measures number of viable cells and cannot determine cell death. Decrease of viable cells can be attributed to cell growth inhibition and/or cell death.

Thank you very much for this comment. We edited our text in accordance with the reviewer's suggestion, lines 289 and 291.

11. Figure 2D&E: If the data shown in the left panels of panels 2D and 2E were from the same primary patient samples, the cell viability data and cell apoptosis data don't match. How could a cell population with 52.7% viable cells is 70% apoptotic?

Thank you very much for this helpful comment.

Figure 2D and 2E data were from the same patients, it has been included the tags REF001 and REF002.

There are two main reasons that make difficult to compare cell population % detected in cell viability and cell apoptosis assays after AML primary cells were treated with EHT1864 and Midostaurin combination: Both assays were done one time in primary cells (no option to calculate standard error) and were used different procedures to analyze cell viability/cell death. The cell viability was calculated by following these steps:

- 1) We calculate an average of three wells that contain medium + CellTiter-GLo, considering our background control.*
- 2) We subtract background control for all measurements.*
- 3) Calculate an average of our control (100% cell viability) and treated wells.*
- 4) Calculate a ratio of treated well/control*100.*

*In the cell death assay, we do not calculate the ratio treatment/control*100 and the controls shows background cell death.*

Here, we want to highlight that the trend observed by the effect of EHT-1864 + Midostaurin combination in AML primary cells is similar in cell viability and cell death assays.

12. Figure 2E: Were these primary patient samples? What do the symbols represent?

We confirm, that these are primary cells. Explanation of symbols/abbreviations were added to the figure legend. Changes to the figure legend are marked in bold.

13. Results, line 251: This subtitle indicates that FLT3-ITD+ cells and FLT3-wt cells were compared.

We thank the reviewer for this cautionary note and we changed the subtitle from "Midostaurin dependent actin polymerization machinery is overexpressed in FLT3-ITD+ cells" to "Midostaurin-dependent actin polymerization machinery is overexpressed in midostaurin-resistant cells, lines 300-301.

14. Figure 3: Densitometry for the blots should be perform.

Thank you very much for your suggestion. We quantified the western blots by ImageJ software and added the information to the blots.

15. Results, lines 324-325: The authors state “BCL-2 and MCL1 were strongly up regulated in MV4-11/MOLM-13 MID-Res compared to their parental cells (Fig. 5A)”. The data in MV4-11 cells supports this statement, while the data in MOLM-13 cells does not. Further, the quality of some the blots shown in Figures 5A and 5C is too poor. Therefore, no meaningful conclusions can be draw based on these blots.

Thank you for this important comment. We corrected the text and replaced figure 5 with an increased resolution version, lines 377-379.

16. There is no animal study data to support that targeting RAC1 is tolerable, especially in combination with midostaurin and venetoclax, and can overcome midostaurin resistance in vivo.

The reviewer rises an important point, that was not addressed in our first version. The development of a FLT3 inhibitors resistant AML xenograft mouse models to show the tolerability of triple combination Midostaurin/EHT1864/Venetoclax is currently planned but data on this are not yet available. We added this important gap of knowledge to our discussion and hope to also spark others people interest with our existing data, line 498.

17. The manuscript will benefit from professional English editing.

The manuscript was edited for proper English language by native English speaking editors at Springer Nature Author Service.

Minor points:

1. Abstract, lines 44-45: please add the word “to” after leads.

We apologize for this typo and corrected it.

2. Introduction, line 77: The abbreviation RAC1 should not be used before the full name of the protein appears in the text.

We changed the text according to the reviewer’s suggestion.

Reviewers' comments:

Reviewer #1 (Remarks to the Author):

In the revised work from Garitano-Trojaola et al the authors have addressed most of the concerns from the reviewer. Nevertheless, there are still two points that need to be revised.

1. Firstly, regarding the effect on cell stiffness and adhesion with midostaurin and Eht1864, the authors only showed the combination of the drugs or midostaurin alone, but the effect of Eht1864 treatment alone was not shown. The authors reasoned that the assays do not allow to compare more than two conditions. If that is the case, please, show in a separate assay the treatment with Eht1864 and with the combination in order to be able to compare both conditions.
2. A minor detail that still needs to be corrected is to indicate that in Figure 2C what is shown is cleaved caspase3 and not caspase 3 total. Please, indicate it in the figure and not only in the figure legend.

Reviewer #2 (Remarks to the Author):

Authors have successfully addressed all of my comments. There was one grammatical change suggestion that authors requested clarification for, however when I looked at their revised, resubmitted manuscript, the sentence I was referring to looked OK. I believe this paper is now ready for publication.

Reviewer #3 (Remarks to the Author):

While the manuscript is very much improved, there remain a few major points that have yet to be adequately addressed.

Major Points:

1. Figure 1B – The quality of the p-FLT3 blots is still very poor.
2. Figure 2C: The authors' response to the high level of baseline dead cells is not acceptable.
3. The authors state "Mcl-1 expression was decreased by midostaurin in MV4-11 and MOLM-13 MID-Sens and MID-Res cells *Fig. 5A; however, Bcl-2 expression was increased by midostaurin in these cell lines (Fig. 5A)." The way this is written implies that the authors are comparing MID-Sens cells treated with and without midostaurin as well as the MID-Res cells treated with and without midostaurin. However, the densitometry numbers comparing Bcl-2 in midostaurin treated cells vs no treatment is only increased in the MID-Sens cells, while in the MID-Res cells, the values are identical. Please edit the text.

Reviewer #1 (Remarks to the Author):

In the revised work from Garitano-Trojaola et al the authors have addressed most of the concerns from the reviewer. Nevertheless, there are still two points that need to be revised.

1. Firstly, regarding the effect on cell stiffness and adhesion with midostaurin and Eht1864, the authors only showed the combination of the drugs or midostaurin alone, but the effect of Eht1864 treatment alone was not shown. The authors reasoned that the assays do not allow to compare more than two conditions. If that is the case, please, show in a separate assay the treatment with Eht1864 and with the combination in order to be able to compare both conditions.

*Regrettably, we are currently limited in access to the cell stiffness measurements as Ana Sancho, who performed the experiments, left Würzburg and moved to San Sebastian. Thus, we were not able to add the recommended analyses. According to the published data of Kunschmann T et al, who demonstrated the effects of RAC1 inhibition on Mouse Embryonic Fibroblast cells (PMID: 31118438), we expect that EHT1864 would also decrease cell stiffness/adhesion forces to Mesenchymal Stromal cells in our AML cells. This result would not affect our main message that MID resistance can be reverted by RAC1 inhibition. Besides, we would like to mention, that the MID-Res cells in our experimental setup will not remain resistant without midostaurin and the comparability of the results obtained would be therefore limited. We hope that this is acceptable to the reviewer and to acknowledge his valuable point we added a cautionary note to figure 4 legend (marked with *).*

2. A minor detail that still needs to be corrected is to indicate that in Figure 2C what is shown is cleaved caspase3 and not caspase 3 total. Please, indicate it in the figure and not only in the figure legend.

We thank the reviewer for this observation. The cleaved caspase3 was added in figure 2C.

Reviewer #2 (Remarks to the Author):

Authors have successfully addressed all of my comments. There was one grammatical change suggestion that authors requested clarification for, however when I looked at their revised, resubmitted manuscript, the sentence I was referring to looked OK. I believe this paper is now ready for publication.

Reviewer #3 (Remarks to the Author):

While the manuscript is very much improved, there remain a few major points that have yet to be adequately addressed.

Major Points:

1. Figure 1B: The quality of the p-FLT3 blots is still very poor.

We apologize for the insufficient figure 1B image quality and replaced it with an increased resolution western blot that includes quantification by ImageJ.

2. Figure 2C: The authors' response to the high level of baseline dead cells is not acceptable.

We thank the reviewer for this helpful comment: We have repeated figure 2C apoptosis assays on recently thawed cell lines, observing lower baseline dead cells.

3. The authors state “Mcl-1 expression was decreased by midostaurin in MV4-11 and MOLM-13 MID-Sens and MID-Res cells *Fig. 5A; however, Bcl-2 expression was increased by midostaurin in these cell lines (Fig. 5A).” The way this is written implies that the authors are comparing MID-

Sens cells treated with and without midostaurin as well as the MID-Res cells treated with and without midostaurin. However, the densitometry numbers comparing Bcl-2 in midostaurin treated cells vs no treatment is only increased in the MID-Sens cells, while in the MID-Res cells, the values are identical. Please edit the text.

According to the reviewer`s suggestion, we modified the text (lines 360-361).

REVIEWERS' COMMENTS:

Reviewer #1 (Remarks to the Author):

This reviewer has been satisfied with the authors corrections and has no further questions. I am therefore happy to accept the paper as it is.

Reviewer #3 (Remarks to the Author):

The authors have adequately addressed my comments.